# Structural insights into the formation and voltage degradation of lithium- and manganese-rich layered oxides

Weibo Hua [1,2], Suning Wang[2], Michael Knapp[1], Steven J. Leake[3], Anatoliy Senyshyn [4], Carsten Richter[3], Murat Yavuz[1], Joachim R. Binder[1], Clare P. Grey[5], Helmut Ehrenberg [1,6], Sylvio Indris [1*] & Björn Schwarz[1]

One major challenge in the field of lithium-ion batteries is to understand the degradation mechanism of high-energy lithium- and manganese-rich layered cathode materials. Although they can deliver 30 % excess capacity compared with today's commercially- used cathodes, the so-called voltage decay has been restricting their practical application. In order to unravel the nature of this phenomenon, we have investigated systematically the structural and compositional dependence of manganese-rich lithium insertion compounds on the lithium content provided during synthesis. Structural, electronic and electrochemical characterizations of $Li_xNi_{0.2}Mn_{0.6}O_y$ with a wide range of lithium contents ($0.00 \leq x \leq 1.52$, $1.07 \leq y < 2.4$) and an analysis of the complexity in the synthesis pathways of monoclinic-layered $Li[Li_{0.2}Ni_{0.2}Mn_{0.6}]O_2$ oxide provide insight into the underlying processes that cause voltage fading in these cathode materials, i.e. transformation of the lithium-rich layered phase to a lithium-poor spinel phase via an intermediate lithium-containing rock-salt phase with release of lithium/oxygen.

[1] Institute for Applied Materials (IAM), Karlsruhe Institute of Technology (KIT), Hermann-von-Helmholtz-Platz 1, 76344 Eggenstein-Leopoldshafen, Germany. [2] College of Chemical Engineering, Sichuan University, No. 24 South Section 1, Yihuan Road, 610065 Chengdu, China. [3] ESRF, The European Synchrotron, 71 Avenue des Martyrs, 38000 Grenoble, France. [4] Heinz Maier-Leibnitz Zentrum, Technische Universität München, Lichtenbergstrasse 1, 85747 Garching, Germany. [5] Department of Chemistry, University of Cambridge, Lensfield Road, Cambridge CB21EW, UK. [6] Department of Geo- and Material Science, Technische Universität Darmstadt, Alarich-Weiss-Strasse 2, 64287 Darmstadt, Germany. *email: sylvio.indris@kit.edu

The ever-increasing demand for electrical energy storage devices, such as electric vehicles poses challenging requirements on long-cycle life, low-cost, and high-energy density cathode materials used in lithium-ion batteries (LIBs)[1,2]. Current state-of-the-art cathodes use polyanionic compounds (e.g., $LiFePO_4$)[3], spinel oxides (e.g., $LiMn_2O_4$)[4,5], or layered lithium transition-metal oxides ($Li[Ni_xMn_yCo_{1-x-y}]O_2$, NMC)[6,7]. One way to reduce the price and increase the capacity limit of layered NMC oxides is via chemical substitution, i.e., the partial replacement of Co and Ni with Li and Mn, aiming at storage of more Li ions in the crystallographic structure[8–10].

These Li- and Mn-rich layered oxides (LMLOs) have improved capacities exceeding 250 mA h g$^{-1}$, much higher than the capacity of conventional cathodes (<200 mA h g$^{-1}$)[11]. The extra capacity was found to be linked to the contribution of reversible oxygen redox activity in LMLOs[12]. Despite that, the commercial application of LMLOs is hindered by their severe voltage decay upon extended cycling. Hence, tremendous efforts have been undertaken to understand the origin of the voltage fade. Several possible mechanisms were proposed, e.g., the formation of $Li_2O$[13,14] or peroxo-like $O_2^{n-}$ ($1 \leq n \leq 3$) dimers[15], the localization of O $2p$ electron holes[16], or the layered-to-spinel phase transition[17,18]. Owing to this complexity, to date, conclusive evidence of the structural and electric details, as well as about thermodynamic phase stability of lithium inserted 3$d$-transition-metal oxides has been lacking.

Herein, we choose Co-free layered $Li[Li_{0.2}Ni_{0.2}Mn_{0.6}]O_2$ as a cathode material because of its high capacity through an interplay of cationic and anionic redox activity, low-cost and resource-friendly nature. The nuances of as-synthesized oxides ($Li_xNi_{0.2}Mn_{0.6}O_y$, $0.00 \leq x \leq 1.52$) are carefully compared and characterized by a combination of analytical methods. The results disclose that cubic spinel phase (space group $Fd\bar{3}m$) is thermodynamically stable at very low Li concentration ($0.00 \leq x < \sim0.40$). Increasing the Li content ($\sim0.40 < x < \sim1.20$) leads to a region with three coexisting phases, i.e., Li-containing spinel ($Fd\bar{3}m$), Li-containing rock-salt-type ($Fm\bar{3}m$) and Li-rich layered phase ($C2/m$). Further incorporation of lithium ions ($\sim1.2 < x < \sim1.52$) into these oxides results in stable monoclinic-layered phases ($C2/m$) with the composition $Li[Li_{0.2+m}(Ni_{0.2}Mn_{0.6})_{1-1.25m}]O_2$. In situ high-temperature synchrotron radiation diffraction (SRD) is used to reveal the dynamics of the spinel-to-layered phase transition. As incorporation of lithium and oxygen into the spinel host matrix occurs, the structural evolution from spinel $(Mn)_{8a,tet}[Ni_{0.75}Mn_{1.25}]_{16d,oct}O_4$ ($Fd\bar{3}m$) to Li-containing rock-salt-type ($Fm\bar{3}m$) and finally to monoclinic-layered $Li[Li_{0.2}Ni_{0.2}Mn_{0.6}]O_2$ phase ($C2/m$) during high-temperature lithiation reaction is uncovered. After ultra-long cycling, the Li-excess layered $Li[Li_{0.2}Ni_{0.2}Mn_{0.6}]O_2$ cathode active material transforms back to Li-containing rock-salt-type ($Fm\bar{3}m$) and then to Li-poor spinel ($Fd\bar{3}m$) phases due to release of lithium and oxygen, which is mainly responsible for the voltage decay.

## Results

### Thermodynamically stable phases of $Li_xNi_{0.2}Mn_{0.6}O_y$ oxides

A series of thermostable oxides ($Li_xNi_{0.2}Mn_{0.6}O_y$, $0.00 \leq x \leq 1.52$) with different contents of lithium and oxygen were obtained by adding different amounts of lithium source ($x$ value) into a hydroxide precursor, details of this synthesis process is in the Supplementary Methods section.

The prepared materials were, respectively, marked as L0.00, L0.08, L0.24, …, L1.52. The Li concentration in the synthesized compounds was somewhat reduced to Li($x - \Delta$) because of the evaporation of Li at high temperature, see chemical composition analysis of selected samples (Supplementary Table 1). The oxygen composition was simultaneously increased ($y$ value) to maintain

the overall electric neutrality of the oxides and to provide more metal coordination sites. The obtained results can be classified according to three regions of provided Li amount: (i) L0.00 to L0.40; (ii) L0.40 to L1.20 and (iii) higher than L1.20, see Fig. 1.

From L0.00 to L0.40, all the reflections in the high-resolution SRD patterns can be indexed according to a single cubic spinel phase (space group $Fd\bar{3}m$), see Fig. 1a. Since neutron powder diffraction (NPD) is able to discriminate elements with similar electronic densities (e.g., Ni and Mn) and to really probe light elements such as Li, high-resolution NPD and SRD were combined to determine the actual structure of L0.00 and L0.40. Simultaneous Rietveld refinement results against NPD and SRD data confirm (Supplementary Figs. 1 and 2) that both L0.00 and L0.40 oxides possess a single spinel phase with the same space group of $Fd\bar{3}m$, while the chemical composition of L0.00 and L0.40 is approximately $(Mn)_{8a,tet}[Ni_{0.75}Mn_{1.25}]_{16d,oct}O_4$ and $(Li)_{8a,tet}[Ni_{0.5}Mn_{1.5}]_{16d,oct}O_4$, respectively. This provides the first direct evidence that Li is inserted into tetrahedral sites of the spinel phase.

To evaluate the fractional occupancy of lithium and transition-metal (TM) atoms on tetrahedral and octahedral sites in the spinel phases for the whole series from L0.00 to L0.40, Rietveld refinements on SRD were completed by assuming a cubic spinel structure with the structural model phase $(Li_xMn_{1-x})_{8a,tet}[Li_yMn_{1-y}]_{16d,oct}O_4$ (Supplementary Fig. 3). With increasing Li concentration, Li atoms have a strong tendency to be localized on 8$a$ tetrahedral sites, and accordingly the occupancy of TM atoms on tetrahedral positions decreases pronouncedly from ~99 % for L0.00 to ~4 % for L0.40 as compared to an almost constant occupancy of TM ions on 16$d$ octahedral sites. Simultaneously, as predicted by the reflections' shift toward higher scattering angles, the lattice parameter of the cubic unit cell is getting smaller, implying the decreased ionic radii of the TM ions with increasing valence state of TM ions and/or the long-range crystallographic disorder-to-order transition in the spinel phase with increasing Li amount. The atomic pair distribution function (PDF) technique was used to reveal directly the interatomic distances of the compounds in real space. As shown in Fig. 1b, TM atoms on both, tetrahedral 8$a$ sites and octahedral 16$d$ sites, are visible in the Li-free cubic spinel phase (L0.0). Importantly, the intensity of $TM_{oct}-(TM/Li)_{tet}$ peak (~3.5 Å) in the PDF analysis is found to decrease as the Li amount increases from L0.00 to L0.40, because of the substitution of TM for Li on tetrahedral sites and thus a reduction of the $TM_{oct}-(TM/Li)_{tet}$ peak occurs, which is consistent with the Rietveld refinement results.

Hard X-ray absorption spectroscopy (XAS) was employed to understand the changes of the bulk oxidation states of the specimens (Fig. 1c). It is clear that the oxidation state of Ni does not change significantly, and remains +2, as evidenced by comparison with a NiO standard. In contrast to Ni, the Mn oxidation state is considerably increased, i.e., it changes from approximately +3 in L0.00 to +4 in L0.40, and its ionic radius reduces significantly ($r_{Mn^{3+}} = 0.58$ Å, $r_{Mn^{4+}} = 0.53$ Å)[19], in good agreement with the shrinking of the cubic unit cell. Generally, the $Mn^{3+}$ state can cause a tetragonal distortion in pure Mn spinel structure ($Mn_3O_4$) due to the Jahn–Teller effect[20,21]. Surprisingly, even though Mn is dominantly in $Mn^{3+}$ state (Fig. 1c), L0.00 is still a cubic spinel phase as a result of the Ni-Mn solid solution. $^7Li$ magic-angle spinning (MAS) nuclear magnetic resonance (NMR) spectroscopy was performed to determine the local structural evolution after high-temperature lithiation of the substances (Fig. 1d). The resonances at 3 ppm can be assigned to Li in diamagnetic compounds like $Li_2CO_3$, while the resonances at 917 and 973 ppm correspond to Li on tetrahedral sites ($LiO_4$) in the high-voltage spinel containing Ni and Mn (i.e., $Li[Ni_{0.5}Mn_{1.5}]O_4$)[22]. The appearance of a broad NMR peak with

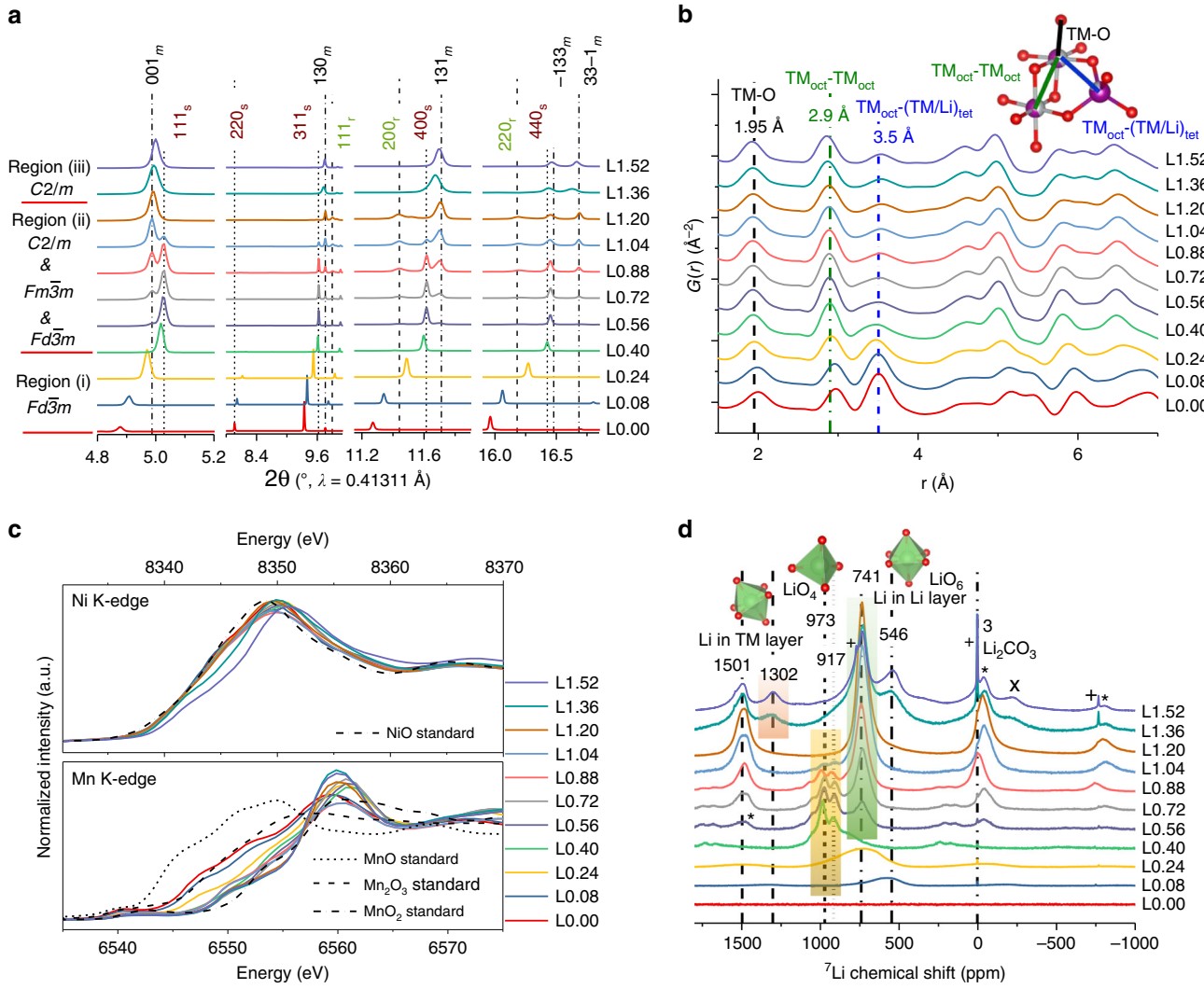

**Fig. 1** Structural and electronic analysis of thermodynamically stable $Li_xNi_{0.2}Mn_{0.6}O_y$ oxides. **a** High-resolution SRD patterns, **b** PDF analysis, **c** Ni K-edge and Mn K-edge XAS spectra, and **d** $^7Li$ MAS-NMR spectra (spinning sidebands are marked with a plus, a cross and an asterisk) obtained after thermal treatment (850 °C for 12 h) of the precursor together with different amounts of $Li_2CO_3$ (L0.00 to L1.52). For the diffraction patterns in Fig. 1a, Rietveld refinements were carried out to estimate the phase composition and lattice parameters of each product, $m$ means layered monoclinic phase, $r$ represents rock-salt-type phase and $s$ means spinel phase. For PDF analysis in Fig. 1b, the first peak at ~1.95 Å (black line) corresponds to the nearest TM-O coordination shell; the second peak at ~2.9 Å (green line) corresponds to the second neighbor coordination via oxygen of two TM ions both located on octahedral sites $TM_{oct} - TM_{oct}$, and the third peak at ~3.5 Å (blue line) to the second neighbor coordination via oxygen with one TM on an octahedral and the other TM or Li on a tetrahedral site $TM_{oct} - (TM/Li)_{tet}$, tet and oct means tetrahedral and octahedral sites, respectively. Each NMR spectrum in Fig. 1d is normalized with respect to the mass of the sample and the number of spectral acquisitions.

such large NMR shifts, different from 3 ppm, indicates that Li is incorporated into a host structure with paramagnetic elements, which is thus different from $Li_2CO_3$. Additionally, the NMR signal is rising and shifting to the resonances at 917 and 973 ppm with increasing Li content, further confirming that Li continuously enters into the tetrahedral sites of the host spinel matrix. The resonance shift is possibly induced by an increase of the Mn oxidation state from around $Mn^{3+}$ (electronic configuration $3d^4$: $t_{2g}^4 e_g^0$) to $Mn^{4+}$ ($3d^3$: $t_{2g}^3 e_g^0$) because more covalent Mn cations could compete more efficiently for binding to nearby oxygen atoms[23]. Combined with the results above, since the overall amount of TM remains constant among these compounds, Li incorporation into the tetrahedral positions of spinel phase would induce more and more Mn ions to be located on the octahedral sites. It is significant to notice that Ni oxidation state maintains a constant value ($Ni^{2+}$, $3d^8$: $t_{2g}^6 e_g^2$), whereas the Mn valence state

increases successively from L0.00 to L0.40, hence, additional oxygen atoms are supposed to be incorporated from atmosphere and/or lithium oxides (e.g., $Li_2O$) into the host structure to maintain electroneutrality and to offer additional octahedral coordination sites for Mn ions that are moved from the tetrahedral sites. Overall, the mean particle size of spinel oxides and the corresponding number of spinel unit cells ($AB_2O_4$) is increased with incorporation of lithium and oxygen (Supplementary Fig. 6), accompanied with oxidation of Mn and atomic redistribution.

From L0.40 to L1.20, all the samples are found to be a mixture of Li-containing spinel phase ($Fd\bar{3}m$), Li-containing rock-salt-type phase ($Fm\bar{3}m$) and Li-containing layered phase ($C2/m$) (Fig. 1a). Rietveld refinements were performed by using a multiple phase model, i.e., spinel $(Li)_{8a,tet}[Ni_{0.5}Mn_{1.5}]_{16d,oct}O_4$, rock-salt-type $[Li_xTM_{1-x}]_{4a,oct}O$ and layered $[Li]_{oct}[Li_{0.2}Ni_{0.2}Mn_{0.6}]_{oct}O_2$, see

Supplementary Fig. 4 and Table 2. As Li concentration increases, the weight fraction of the spinel phase gradually diminishes from ~100 % for L0.40 to ~5 % for L1.20, while the percentage of layered phase and Li-containing rock-salt-type phase increases to ~82 and ~12%, respectively, illustrating that the spinel phase transforms progressively to the layered phase accompanied by a formation of a Li-containing rock-salt-type phase. No obvious change in lattice parameters is found during phase transition pointing to a predominant two-phase transformation mechanism. When more Li ions are accommodated into the spinel structure, the Li ions on tetrahedral sites and TM ions on octahedral sites could move randomly into the remaining empty octahedra (16c) and the other octahedra (16d) to yield a rock-salt structure $[Li_{2x}TM_{2-2x}]_{16c,oct}[Li_{2x}TM_{2-2x}]_{16d,oct}O_4$ (i.e., $[Li_xTM_{1-x}]_{4a,oct}O$) in which the original spinel cubic-close packed (ccp) oxygen framework is maintained[24]. The typical layered rhombohedral structure ($R\bar{3}m$) can be considered as a Li-containing ordered rock-salt derivative, where octahedrally coordinated TM and Li ions perfectly form alternating layers confined to the $(111)_r$ planes of ccp oxygen lattice[25]. PDF analysis (Fig. 1b) displays that TM cations are mainly located on octahedral sites for these samples with different phases. Only slight variations of the PDF signal from L0.40 to L1.20 demonstrate a very similar local structure in spinel, rock-salt-type and layered phase because all these structures have a ccp oxygen lattice. XAS results (Fig. 1c) show that oxidation states of Ni and Mn for these materials remain the same as in the L0.40 sample, all are assigned to $Ni^{2+}$ and $Mn^{4+}$, respectively. Figure 1d shows that the NMR peaks at 917 and 973 ppm representing Li in the typical high-voltage spinel phase decrease gradually, whereas the resonance at 741 ppm corresponding to Li on octahedral sites in the Li layer in the typical layered oxides[22,26] increases significantly. The results clearly prove that more and more Li ions tend to be located on octahedral sites forming the layered and/or rock-salt structure with Li atoms further incorporated into the host structure. Altogether, owing to the fact that the oxidation state of Ni and Mn does not change considerably from L0.40 to L1.20, the phase transformation induced by chemical lithiation, from spinel to rock-salt-type respectively to layered phase, is supposed to be accompanied by an oxygen uptake, cationic rearrangement and surface reconstruction (accommodating more lithium and oxygen).

When the provided Li content is higher than that of L1.20, both spinel and rock-salt-type phases have been completely converted into the layered phase, as shown by the evolution of diffraction patterns in Fig. 1a. Most of the reflections in diffraction patterns for both L1.36 and L1.52 can be indexed to a rhombohedral phase ($R\bar{3}m$) and a few weak reflections can be ascribed to the superstructures corresponding to a monoclinic symmetry ($C2/m$). Indeed, both rhombohedral phase and monoclinic phase possess an O3-type structure. Unlike the rhombohedral structure ($R\bar{3}m$), where the majority of transition-metal cations are randomly located at the TM layer, a honeycomb ordered monoclinic-layered structure ($C2/m$) could accommodate the excess Li ions within the TM layer forming the in-plane Li/TM ordering (i.e., superlattices). As all the reflections in the SRD patterns of the samples could be well indexed according to a single monoclinic phase ($C2/m$), Rietveld refinements were performed by assuming a layered structural model $[Li]_{oct}[Li_{0.2}Ni_{0.2}Mn_{0.6}]_{oct}O_2$, see Supplementary Fig. 5 and Table 2. The lattice parameters of L1.52 are slightly smaller than those of L1.36, as observed by a shift of reflections to higher 2-θ angles, revealing that the TM ions in L1.52 are further oxidized. There is no considerable variation in PDF signal for these Li-excess compounds (Fig. 1b). XAS results show that the Mn valence does not change perceptibly, while the Ni valence at very high-Li content L1.52 is slightly increased (Fig. 1c), again proving a continuous oxygen uptake during chemical insertion of Li ions

into layered structure. More inspiringly, the new resonances observed at around 1302 and 1501 ppm in NMR spectra, corresponding to Li in the TM layer (octahedral coordination) typical for $Li[Li_{1/3}Mn_{2/3}]O_2$ phase[22], suggest that the excess of Li ions transfers to the TM layer in layered phase forming the in-plane superlattice, which matches precisely with the results obtained from the diffraction experiments of these samples. As the content of TM ions is fixed within the layered and rock-salt-type phase, the equally distributed TM and Li ions within the layers in rock-salt structure separate into a Li layer (only few TM ions) and a TM layer that forms a superstructure together with considerable amount of Li (~20 %) also present in this layer, i.e., a fully disordered structure transforms into a more ordered one. As a consequence, Li incorporation into the layered structure is also supposed to be followed by an oxygen adsorption and an increase in the particle size of layered oxides.

This section describes a portion of a $Li_xNi_{0.2}Mn_{0.6}O_y$ oxide phase diagram (grand potential), revealing the tendency of the Mn-rich oxides towards formation of different phases, depending on the Li content in the system, during synthesis and discharge-charge process of LMLOs. These new findings not only provide the prerequisites for synthesis of lithium insertion compounds with high performance, i.e., the precise preparation of monoclinic-layered $Li[Li_{0.2}Ni_{0.2}Mn_{0.6}]O_2$ oxide (see Fig. 2 and Supplementary Fig. 7), but also help us understand the structural degradation of these cathode materials during long-term cycling (Fig. 3).

**Electrochemical properties of optimized layered Li $[Li_{0.2}Ni_{0.2}Mn_{0.6}]O_2$ cathode.** Li- and Mn-rich layered compounds with the nominal formula $Li_{1.2}Ni_{0.2}Mn_{0.6}O_2$ were prepared by controlling the synthesis process precisely and offering an appropriate amount of Li (L1.28, see experimental part). A combination of SRD, NPD, and elemental analysis was used to determine the real structure and chemical composition of the as-synthesized compound, see Fig. 2a, b and Supplementary Table 1. These results indicate that the L1.28 oxide is composed of a layered monoclinic phase (space group $C2/m$) with the desired composition. Simultaneous Rietveld refinement against SRD and NPD patterns yield lattice parameters of $a = 4.9579(2)$ Å, $b = 8.5816(2)$ Å, $c = 5.0324(2)$ Å and $\beta = 109.25(2)°$, and a reliable structure model of $[Li_{0.96}Ni_{0.04}]_{oct}[Li_{0.24}Ni_{0.16}Mn_{0.6}]_{oct}O_2$ (see Supplementary Table 3). Scanning electron microscopy (SEM, Supplementary Fig. 6) images show that the primary particle size of L1.28 is 100–300 nm. These small crystalline grains with platelet-like shape tend to agglomerate and form large secondary particles. XAS spectra (Fig. 3b) reveal that the oxidation states are mainly determined to be 2 + and 4 + for Ni and Mn, respectively.

Galvanostatic charge and discharge tests of a L1.28 electrode at a current density of 32 mA g$^{-1}$ show a long-term cycling performance (over >1 year) in a lithium battery (CR2032-type coin cells), see left part in Fig. 2c. There is a fast capacity fading from ~240 mA h g$^{-1}$ to ~210 mA h g$^{-1}$ during the initial 50 cycles (stage I), which probably results from the isolation of active electrode material from the conductive composites, due to the anisotropic lattice expansion/contraction of electrode particles during cycling (see in situ SRD results, Supplementary Fig. 18) and the formation of solid electrolyte interphase (SEI) films on the cathode surface[27–29]. The discharge capacity of the L1.28 cathode is about 195 mAh g$^{-1}$ after 300 cycles, maintaining around 80 % of the initial capacity (stage II). Only changes in the Ni valence state ($Ni^{2+} \rightleftharpoons Ni^4 + 2e^-$) are observed during the charge and discharge process, see in situ XAS results (Supplementary Figs. 20 and 21), whereas the oxidation state of $Mn^{4+}$ remains constant demonstrating that tetravalent Mn in

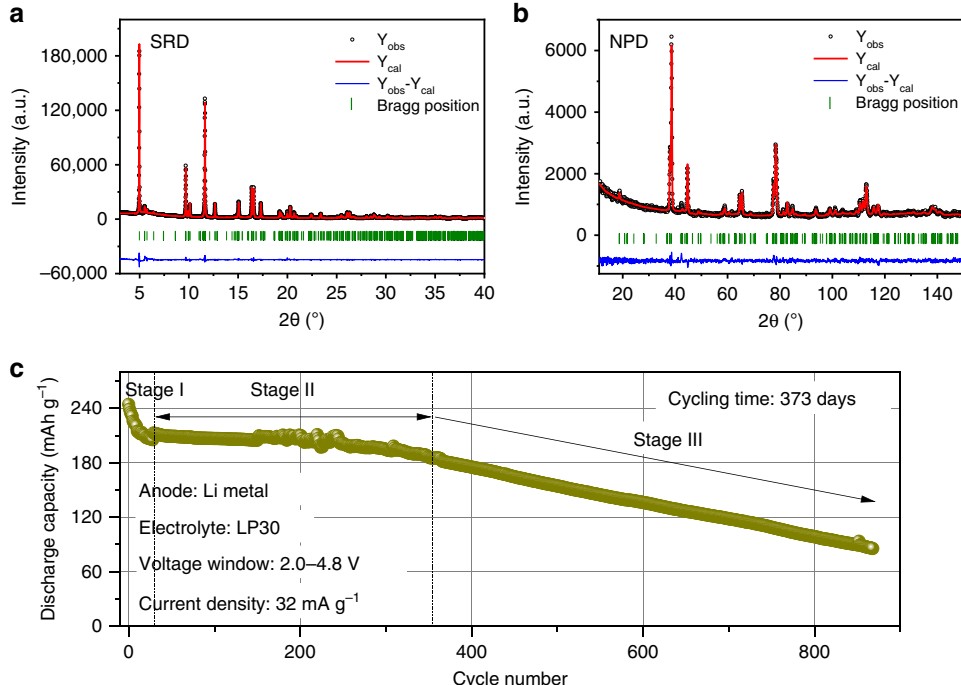

**Fig. 2** Obtained monoclinic-layered Li[Li$_{0.2}$Ni$_{0.2}$Mn$_{0.6}$]O$_2$ cathode showing long-term degradation process. Simultaneous Rietveld refinement on **a** SRD ($\lambda = 0.41231$ Å) and **b** NPD ($\lambda = 1.54825$ Å) patterns of Li$_{1.2}$Ni$_{0.2}$Mn$_{0.6}$O$_2$ (L1.28); **c** electrochemical cycling performance of L1.28 electrode vs. Li metal between 2.0 and 4.8 V at room temperature.

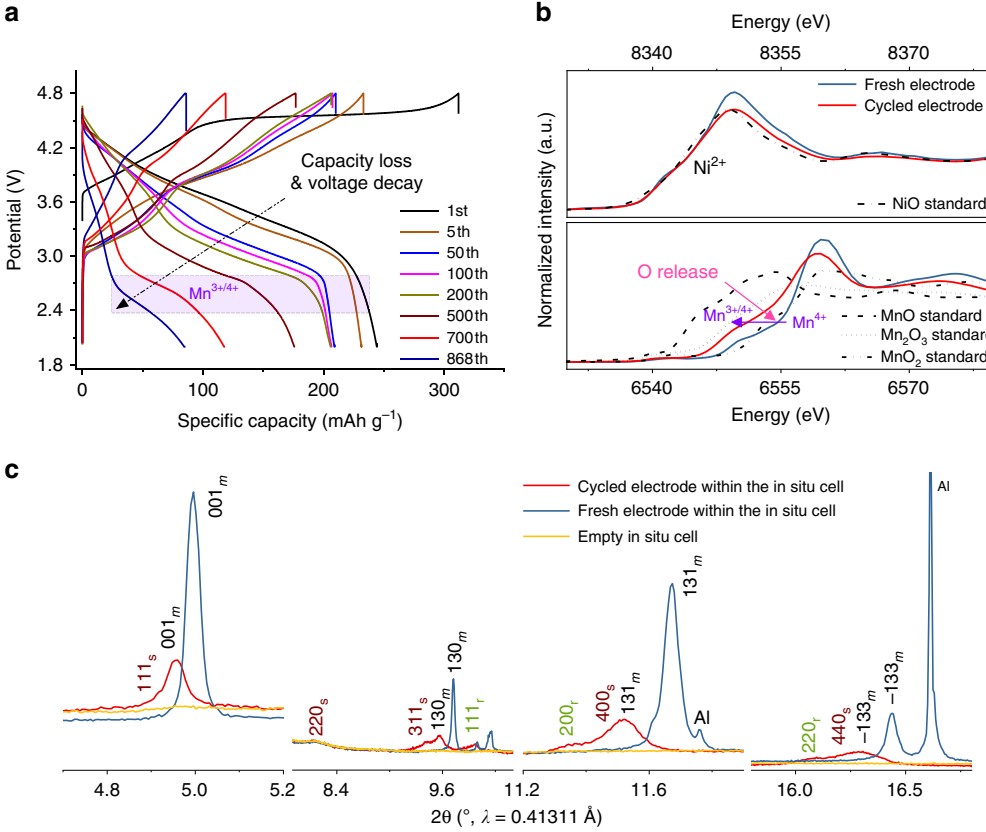

**Fig. 3** Structural degradation of Li[Li$_{0.2}$Ni$_{0.2}$Mn$_{0.6}$]O$_2$ (L1.28) cathode after long-term cycling. **a** Selected charge and discharge curves of a L1.28/Li cell during prolonged cycling; **b** Ni K-edge and Mn K-edge XAS spectra of the fresh and the cycled L1.28 electrode (after 868 cycles) measured within an in situ coin cell, respectively; **c** their corresponding SRD patterns, and the SRD pattern of the empty in situ coin cell as a reference.

octahedral coordination cannot provide extra electrons[16]. Such a good durability (within the first 300 cycles) provides a clear evidence that the oxygen redox process is at least partially reversible within a relatively long period, contributing to a specific capacity of ~100 mA h g$^{-1}$, because the Ni$^{2+}$/Ni$^{4+}$ redox activity can only compensate for 0.4 Li ion extraction from Li$_{1.2}$Ni$_{0.2}$Mn$_{0.6}$O$_2$ corresponding to the remaining capacity of ~100 mA h g$^{-1}$. After 300 cycles (stage III), the discharge capacity of L1.28 cathode decreases linearly with an increasing number of cycles, pointing out that the Li-excess layered structure after the gradual oxygen loss has a tendency to form a thermodynamically stable equilibrium state, i.e., Li-rich layered and Li-containing rock-salt-type and Li-poor spinel coherent phases, during non-equilibrium (de)lithiation (see Fig. 3), as predicted from the thermodynamic equilibrium state of Li$_x$Ni$_{0.2}$Mn$_{0.6}$O$_y$ (Fig. 1).

**Why and how do Li-rich layered cathode materials degrade?**. Selected charge-discharge curves of the L1.28/Li cell are displayed in Fig. 3a, showing a severe voltage decay and capacity loss i. The initial charge profile of the L1.28 cathode exhibits a monotonically increasing region below 4.5 V and a high-voltage plateau at about 4.5 V vs. Li/Li$^+$, which is ascribed to the oxidation of Ni$^{2+}$ to Ni$^{4+}$ and the successive activation process, respectively. The subsequent charge profiles show a different characteristic. This is probably related to the generation of oxygen vacancies or phase transitions after activation[30], see in situ SRD and XAS results of L1.28 electrode (Supplementary Figs. 18–21). Both capacity and voltage profile of L1.28 cathode decrease steadily during the first 50 cycles (stage I in Fig. 2c), indicating an increased internal resistance, i.e., slow Li-ion diffusion, in the electrode and/or a small amount of irreversible lithium loss from the layered host structure[31,32]. Obviously, the capacity retention of the L1.28 cathode is almost 100 % from the 50th to the 200th cycles, but a clear plateau located at around 2.7 V is found in the discharge curve of the 200th cycle. This plateau can be ascribed to the lower-voltage Mn$^{3+}$/Mn$^{4+}$ redox couple, as confirmed in previous studies[33]. We note that the initial capacity loss of about 30 mA h g$^{-1}$ is not always observed in the first 50 cycles (Supplementary Fig. 7), but it is the good capacity retention with serious voltage decay (stage II) that is common to Li-rich layered cathodes within a limited cycling time (e.g., 300 cycles)[34,35]. This kind of voltage fade is primarily ascribed to the layered-to-spinel phase transition in the lattice structure of LMLO electrodes[13,16,36,37]. Up to now, however, there is no direct experimental observation of the formation of spinel phase (AB$_2$O$_4$, Fd$\bar{3}$m, low Li/O content) at stage II. Instead, a spinel-like phase (Fd$\bar{3}$m), in which all the cations are located on octahedral sites in the crystal structure, was proposed to explain the structure of a surface reconstruction layer in numerous publications[34,38]. For instance, a possible phase transformation pathway, i.e., from a layered to a LT-LiCoO$_2$ type defect spinel-like (Fd$\bar{3}$m) to a disordered rock-salt phase (Fm$\bar{3}$m), was described by Zheng et al[35]. during the first 100 cycles. Such a reaction would involve a huge amount of Li/O release during the phase transition from Li-rich layered (Li$_{1.2}$Ni$_{0.2}$Mn$_{0.6}$O$_2$) to Li-free rock-salt (TMO, Fm$\bar{3}$m) structure. Therefore, the fundamental questions in the field of Li-ion batteries that need to be clarified are (1) what is the nature of this so-called spinel-like (Fd$\bar{3}$m) phase and (2) whether there is a rate-limiting step before the formation of spinel, AB$_2$O$_4$, phase (Fd$\bar{3}$m) during cycling.

After 50 and 200 cycles, the reflections in the SRD patterns of the cycled L1.28 electrodes can be indexed to a layered structure with space symmetry of C2/m and/or R$\bar{3}$m, see Supplementary Fig. 9a. Compared to the SRD pattern of L1.28 electrode before cycling, all reflections in the SRD pattern of L1.28 cathode after 200 cycles move to lower 2-θ angles (lattice expansion) and the

split reflections of $\bar{1}33_m$/$33\bar{1}_m$ or $018_h$/$110_h$ tend to merge into a single reflection (less distortion from a cubic phase). The oxidation state of TM in the cycled electrode does not change substantially and can still be assigned to Ni$^{2+}$ and Mn$^{4+}$ (Supplementary Fig. 9c). Thus, it would be reasonable to speculate that the expansion of the Li-rich layered unit cell is caused by the formation of more disordered [Li/TM]O$_6$ octahedra, i.e., from Li-O-[Li/TM] chains in the ordered layered structure to partially [Li/TM]-O-[Li/TM] linkages in the disordered rock-salt structure, which is closely related to the lattice contraction behavior in the fully disordered Li-containing rock-salt structure during synthesis of LMLOs in Fig. 4. Additionally, the TM occupancy on the Li sites in the layered structure of L1.28 electrode has increased from ~3(2) % before cycling to ~6(2) % after 200 cycles (Rietveld refinement results in Supplementary Fig. 9b and Table 4), again suggesting the generation of partially disordered Li-containing rock-salt-type phase (Fm$\bar{3}$m). On the other hand, lack of evidence for the spinel phase (AB$_2$O$_4$, Fd$\bar{3}$m) formation at this stage indicates that the cycled L1.28 electrode dose not reach equilibrium within five months of cycling due to a small amount of irreversible Li/O loss (see Fig. 1). Therefore, the commonly called "spinel-like" phase formed within a limited period can be interpreted as the accumulation of layered (C2/m or R$\bar{3}$m) and Li-containing rock-salt [Li$_x$TM$_{1-x}$]O (Fm$\bar{3}$m) nano-domains in our electrodes, which agrees quite well with the analysis of activation process (layered-to-rock-salt transition) of L1.20 electrode during the first 100 cycles in Supplementary Figs. 7 and 8. The disordered rock-salt-type phase can block the lithium diffusion pathway to some extent and thus result in a worsening electrochemical performance[39].

After over 1-year cycling (stage III in Fig. 2c), the Mn$^{3+}$/Mn$^{4+}$ redox couple at ~2.7 V has become more pronounced, accompanied with significant capacity loss. It is obvious that the Ni valence state of cycled L1.28 electrode remains the same as in the pristine electrode (Ni$^{2+}$), but the Mn oxidation state is reduced considerably from +4 for the fresh electrode to +3/+4 for the cycled electrode, see Fig. 3b, giving indirect evidence for oxygen release from the ccp oxygen lattice. The SRD patterns of the fresh and cycled electrode after 868 cycles show the long-range structural rearrangement after long-term testing. The main reflections in the SRD pattern of the cycled electrode can still be indexed to the layered phase, while these reflections shift toward lower scattering angles when compared with the SRD pattern of the pristine electrode, which coincides with the reduction of Mn valence state resulting in larger unit cell parameters of the layered phase. Significantly, the weak reflections in SRD pattern of the cycled L1.28 electrode are only partially indexed to the rock-salt-type phase such as the 200$_r$ and 220$_r$ reflections, some of them can be unambiguously indexed to the spinel phase (e.g., 311$_s$). These analyses are also compatible with the ex situ SRD results in Fig. 1a, the in situ SRD analysis of the L0.88 electrode with coherent layered and spinel, and rock-salt-type phases during cycling (Supplementary Figs. 16 and 17) and the in situ high-temperature SRD results during the formation of LMLOs (see Fig. 4). Moreover, the broad Bragg reflections of the cycled electrode indicate distortions in the long-range order, i.e., a generation of dislocations/strain after repeated Li insertion/extraction[40], as shown in Fig. 5. Taken together, thermodynamically driven formation of the Li-poor spinel phase (AB$_2$O$_4$, Fd$\bar{3}$m) is finally observed after ultra-long cycling time, which arises from irreversible release of lithium and oxygen from layered crystal lattice and is supposed to be located at the surface (see Fig. 5). Unfortunately, the accumulation of layered and rock-salt-type, and spinel domains with a large lattice mismatch could induce a much higher diffusion barrier for Li ions, compared to the stage II, and thereby a considerably capacity loss at stage III in Fig. 2c. In

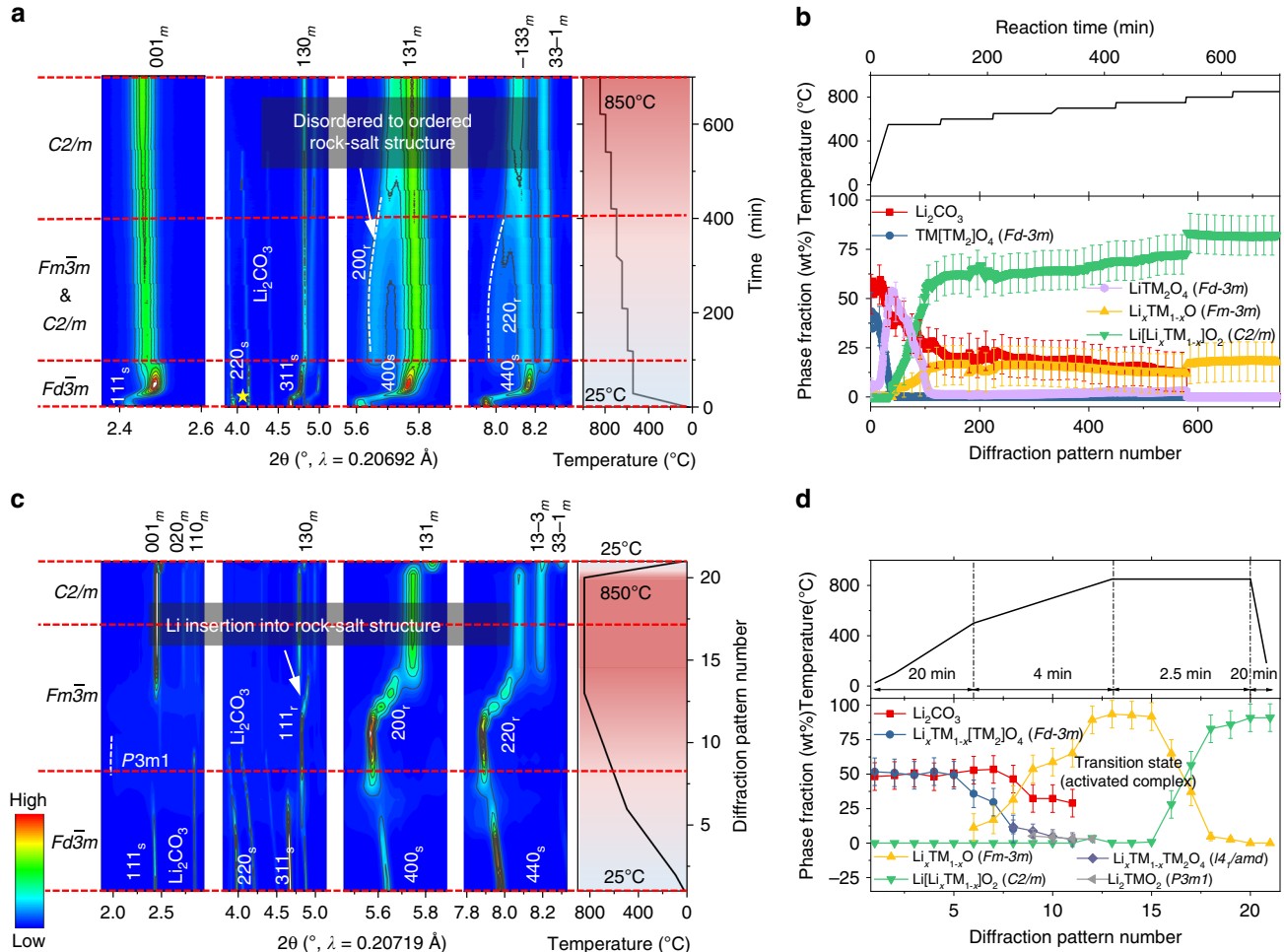

**Fig. 4** The original formation mechanism of LMLOs and the anomalous lattice contraction behavior in the cubic Li-containing rock-salt-type phase. Contour plot of in situ HTSRD pattern of a mixture of L0.00 and $Li_2CO_3$ under **a** slow-heating and **c** fast-heating process, respectively. On the right side of the SRD pattern is the corresponding calcination process. Each diffraction pattern was analyzed using the Rietveld refinement method, the resultant weight fraction of various phases showing the Li/O-incorporation-induced structural evolution during **b** slow and **d** fast-heating route between 25 and 850 °C.

addition, the capacity fading is probably also a consequence of the slow dissolution of manganese (III) ions in the spinel phase ($2Mn^{3+} \rightarrow Mn^{4+} + Mn^{2+}$), the lithium dendrite and SEI growth in the coin cell, and electrolyte evaporation[41–44]. Most importantly, such long-term cycling battery test could give valuable information on the kinetic behavior of a phase transition from a Li-rich layered $Li[Li_xTM_{1-x}]O_2$ ($C2/m$ and/or $R\bar{3}m$) to a partially disordered Li-containing rock-salt $[Li_xTM_{1-x}]O$ ($Fm\bar{3}m$) and then to a Li-poor spinel ($Li_xTM_{1-x}$)$[TM_2]O_4$ ($Fd\bar{3}m$) structure.

To better understand the kinetics of the degradation reaction, we map the three-dimensional (3D) strain ($\Delta d/d$) field inside a single crystallite of the cycled L1.28 cathode (after 868 cycles) with Bragg coherent diffractive imaging (BCDI). Details of the experimental method and data reduction can be found in the supplementary material. Figure 5 shows a distinct homogeneous tensile strain towards the surface of the crystallite from its core, i.e., the lattice parameter is increasing. The average $d_{001_m}$ lattice parameter of the crystal determined by BCDI from the layered structure is 0.4805(2) nm (zero-strain in Fig. 5) and is very close to the corresponding value determined by SRD in Fig. 3c (0.4802 (2) nm). BCDI cannot distinguish unambiguously if the structure of the crystal is layered or spinel, but in this case BCDI is sensitive to both phases because the $111_s$ spinel would interfere with the $001_m$ layered reflection. However, given the XRD results from many millions of crystallites we anticipate Li-poor spinel on the

very surface with a larger lattice parameter than the layered phase (L1.28, 0.4751(2) nm) and spinel $LiNi_{0.5}Mn_{1.5}O_4$ (L0.40, 0.4725 (2) nm), consistent with the strain distribution observed in Fig. 5. The Li-poor spinel layer is probably thinner than the spatial revolution (35 nm), see Supplementary Fig. 12, of the reconstructed object image but the distortion of the lattice between the two phases extends into the crystallite, and is therefore visible. Hence, the degradation mechanism of Li-rich layered cathodes can be thought of a surface-bulk-limited reaction (see Supplementary Movie 1 and 2).

**Origin of oxygen-incorporation during synthesis of Li-rich oxides.** Bearing in mind that Li-rich layered phase would convert into Li-poor spinel phase after ultra-long cycling time, it is therefore interesting to investigate how the structural evolution from Li-free spinel to Li-rich layered phase occurs during synthesis. Herein, the Li-free spinel L0.00 ($Fd\bar{3}m$) was mixed with a desired amount of $Li_2CO_3$ as starting materials to study the lithiation mechanism and to determine the rate-limiting step during spinel-to-layered phase transition, as shown in Fig. 4.

In situ high-temperature SRD (HTSRD) was firstly carried out to decipher the dynamics of the chemical reaction between Li-free spinel oxide and Li/O species during multi-step slow calcination process. The intensity of the $220_s$ reflection is extremely sensitive to the occupancy of the $8a$ (tetrahedral) site by TM cations, the

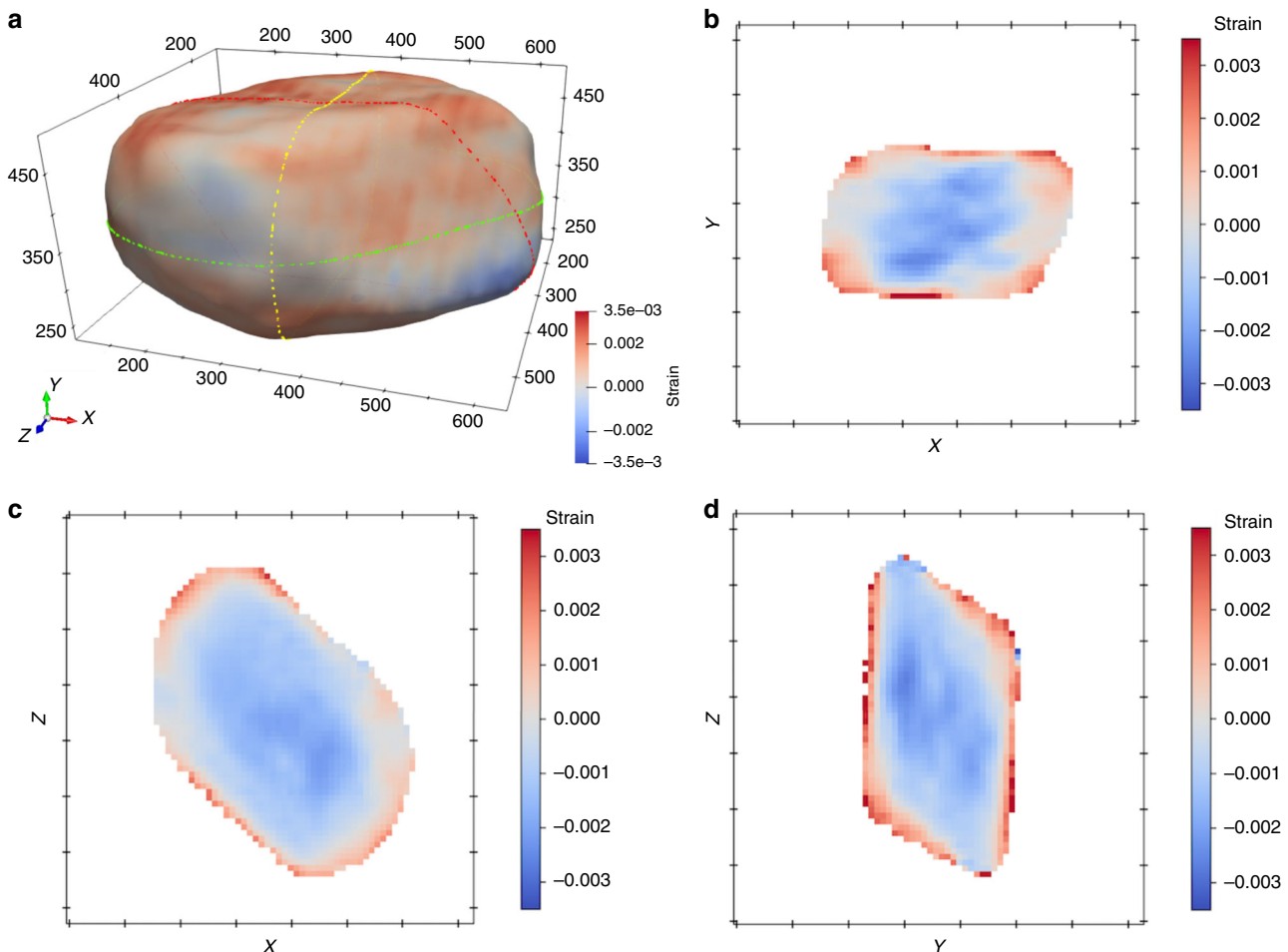

**Fig. 5** Bragg coherent diffractive imaging reveals the internal structure of a single crystallite from the L1.28 cathode after 1-year cycling. **a** Three-dimensional render of the crystallite, where the color map denotes strain and corresponding cross sections through the 3D structure in the **b** XY, **c** XZ, and **d** YZ planes. Tick spacings along all axes correspond to 100 nm.

intensity ratio of the reflection 311$_s$ and 111$_s$ is an alternative indicator for estimating the degree of TM cation mixing between tetrahedral and octahedral positions (the ratio reduces with decreasing occupation of TM on the tetrahedral sites) in the spinel structure[45]. For the multi-step slow calcination procedure (Fig. 4a), as the temperature increases to 550 °C, the intensity of the 220$_s$ reflection in the SRD diffraction pattern of L0.00 diminishes gradually together with the decreasing intensity ratio of the 311$_s$/111$_s$ reflections. The Rietveld refinements results (Fig. 4b) show that the weight fraction of the Li-containing spinel phase (similar to Li[Ni$_{0.5}$Mn$_{1.5}$]O$_4$) is increased at the expense of the Li-free spinel phase, suggesting that lithium can easily enter into the tetrahedral sites and push Mn ions to be located at 16$c$ positions forming the more ordered Li-containing spinel phase. Synchronously, the inserted oxygen atoms cannot only increase the oxidation state of Mn, but offer more octahedral coordination sites for the transferred TM cations in the spinel matrix. In addition, the lithiation reaction at this stage accelerates the decomposition of Li$_2$CO$_3$ (melting point: 723 °C), revealing that the spinel phase exhibits a good reaction activity for the lithiation reaction during the heat treatment. With an increase of temperature from 550 to 600 °C, the phase transition from cubic Li-containing spinel phase to Li-containing rock-salt-type phase and Li-containing layered phase is pronouncedly detected. During this phase transformation, the weight fraction of the Li-containing spinel phase gradually decreases from ~82 to ~2 %,

while the relative percentage of the layered phase and the rock-salt-type phase raises to ~80 and ~18 %, respectively, see Fig. 4b. When the temperature is higher than 600 °C, the decreased fraction of the Li-containing rock-salt-type phase and the increased percentage of the Li-rich layered phase reveal a successive transformation from rock-salt-type to layered phase. Interestingly, the $a$ parameter of the cubic rock-salt unit cell reduces from ~4.1815(2) to ~4.1452(2) Å when the temperature is increased from 550 °C up to ~750 °C, as observed from the reflections shifting to higher scattering angles, demonstrating a strong lattice shrinkage in the Li-containing rock-salt-type phase. This unusual phenomenon can be explained by the phase transition from disordered rock-salt to ordered rock-salt structure (i.e., layered phase) because a random distribution of the larger Li-ions with the TM ions requires more space than the alternating layers of Li-ions and TM ions ($r_{Li^+}$ = 0.76 Å, $r_{Ni^{2+}}$ = 0.69 Å, $r_{Mn^{4+}}$ = 0.53 Å)[19]. The reaction of disordered rock-salt-type phase with Li/O species occurs very slowly at high temperature, implying that the transformation from disordered rock-salt to ordered layered structure is kinetically hindered, which might be related to the voltage degradation of LMLOs (layered to partially rock-salt transition) within a relatively long period (e.g., 200 cycles), see Fig. 3 and Supplementary Fig. 9. A low heating rate and a long residence time are sufficient for lithiation reaction to achieve equilibrium. The portion of oxide products in Fig. 4b is mainly determined by thermodynamics, i.e., the standard Gibbs energies

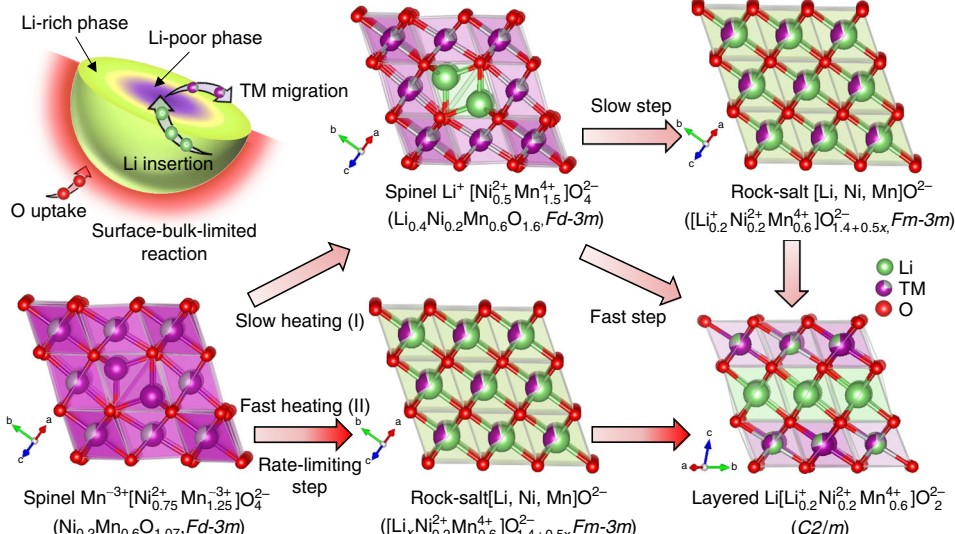

**Fig. 6** Two distinct pathways of Li-rich layered oxide formation from Li-free spinel oxide. Schematic diagram of lithium- and oxygen-incorporation-induced structural and electrical evolution during synthesis of layered $Li[Li_{1.2}Ni_{0.2}Mn_{0.6}]O_2$ cathode materials, revealing the formation mechanisms of Mn-rich cathodes during slow (I) and fast (II) thermal treatments, and the possible origin of voltage decay in high-energy lithium-excess layered oxides, i.e., a kind of partially reversed formation pathway (II) of LMLOs under non-equilibrium conditions (lithium extraction accompanying oxygen evolution).

of all the reactants and products, in excellent agreement with the analysis of $Li_xNi_{0.2}Mn_{0.6}O_y$ oxides in Fig. 1.

Considering that the structural evolution of Li-rich layered oxides during electrochemical cycling is far away from thermal equilibrium (see Fig. 3), a fast thermal treatment process was then applied to gain new insights into the kinetically controlled non-equilibrium lithiation reaction. In the early stage of the reaction, the Li-free spinel phase rapidly converts to the intermediate phases, i.e., layered $Li_2TMO_2$ phase ($P3m1$), spinel phases ($Fd\bar{3}m$ and $I4_1/amd$) and rock-salt-type phase ($Fm\bar{3}m$), see Fig. 4c, d and Supplementary Table 5. The formation of over-lithiated 1T-$Li_2TMO_2$ phase can be ascribed to the high-Li concentration at the surface of the crystallites, implying the Li redistribution in the non-equilibrium phase transition. All these intermediate phases transform into Li-containing rock-salt-type phase at around 800 °C and finally become the layered phase with a further increase of temperature at about 850 °C, revealing that the lithiation is frustrated by the high activation energy of the phase transition from Li-containing rock-salt-type phase to layered phase. Specifically, it is well known that the $(200)_r$ plane of ionic rock-salt-type oxides can be regarded as a non-polar cleavage plane. In contrast, the $(111)_r$ plane is polar and the corresponding phase unstable[46,47]. The results of X-ray diffraction simulation (Supplementary Fig. 24) show that the $200_r$ reflection is the strongest in the random rock-salt-type $[Li_xNi_{1-x}]_{oct}O$ oxide, and the intensity of the $111_r$ reflection is reduced with more lithium intercalation into the rock-salt structure. We use this information, i.e., the relative intensity of the $111_r$ and $200_r$ reflections ($I_{111}/I_{200}$), as a means to assess the lithium content in the Li-containing rock-salt-type phase. As shown in Fig. 4c and Supplementary Table 4, $I_{111}/I_{200}$ ratio drops progressively with increased reaction time, revealing that lithium ions are continuously inserted into the rock-salt-type phase. More interestingly, the reflections of this cubic rock-salt-type phase initially shift toward lower two-θ scattering angles and obviously move to higher angles as the reaction proceeds, in good agreement with the contraction of the Li-containing rock-salt unit cell in Fig. 4a. This may be due to the $[Li/TM]O_6$ octahedra becoming more regular, i.e., from [Li/TM]-O-[Li/TM] linkages in the rock-salt structure to Li-O-[Li/TM] chains in the monoclinic-layered structure. Noticeably, the

monoclinic-layered structure is maintained after cooling to room temperature, indicating that lithium and oxygen insertion into the spinel/rock-salt-type phase results in a stable layered phase. These data unambiguously demonstrate that the non-equilibrium phase transition from Li-free spinel to Li-rich layered structure is rate-limited by the formation of an often ignored Li-rich rock-salt-type intermediate. We interpret the overall results throughout the paper as the ordering/disordering of $[Li/TM]O_6$ octahedra (i.e., cation mixing) that occurs in the whole crystal structure with O-incorporation/release into/from the ccp oxygen lattice at the surface during non-equilibrium (de)lithiation process. It is notable that the Li/TM cation disorder was believed to be another major contribution to the voltage fade behavior in Li-rich layered cathodes during cycling at room temperature[15,34].

## Discussion

On the basis of all results above, two possible lithiation reaction mechanisms with oxygen-incorporation are proposed, as shown in Fig. 6. The uncovered phase transition during synthesis reveals that the formation mechanism of LMLOs is a surface-bulk-limited reaction. Since all the spinel, rock-salt-type and layered phases have the common ccp oxygen lattice[48], oxygen atoms are supposed to be only added to the oxygen surface lattice, thus offering the possibility of crystal growth or recrystallization. The fast-heating procedure results in a non-equilibrium reaction pathway. During synthesis of layered $Li[Li_{0.2}Ni_{0.2}Mn_{0.6}]O_2$ oxide under rapid-heating conditions, the cubic spinel $Mn[Ni_{0.75}Mn_{1.25}]O_4$ phase (L0.00) experiences a transformation from a Li-free spinel ($Fd\bar{3}m$) to a fully disordered Li-containing rock-salt $Fm\bar{3}m$ since Li/O are incorporated and the system does not have enough time for cation ordering. Subsequently, a Li-rich layered structure ($C2/m$) is formed with the incorporation of lithium and oxygen. As a gradual Li/O loss occurs during long-term cycling, the Li-rich layered $Li[Li_{0.2}Ni_{0.2}Mn_{0.6}]O_2$ cathode tends to transform back to the "Li-poor" spinel phase ($Fd\bar{3}m$), but this transition is limited by the generation of the Li-containing rock-salt phase ($Fm\bar{3}m$). Such an inversed relationship between formation pathway and degradation process of LMLOs is not a coincidence but intrinsic due to the fact that both occurs under non-equilibrium conditions driven by Li/O concentration

gradient. The activation energy barrier for the transition from Li-containing rock-salt intermediate to Li-rich layered oxide can easily be transcended at elevated temperatures, but such an activation barrier could impede the thermodynamic phase transformation at room temperature. This could help to explain why the TM cation migration into the octahedral positions of Li-containing rock-salt-type/layered phase in the near-surface region, rather than the tetrahedral positions of cubic spinel phase ($Fd\bar{3}m$), is so often observed in Li-rich layered cathode materials during a short-term to midterm (e.g., 100 cycles) cycling (kinetic control) and why Li-rich layered oxides have the tendency to form Li-containing rock-salt-type and Li-poor spinel coherent phases after a certain amount of Li/O loss during prolonged cycling (>500 cycles) at room temperature (i.e., thermodynamically driven)[34,35,49,50]. Finally, we believe that these findings will enable a new and comprehensive look into the interplay of lithium and oxygen during lithium insertion/extraction, and help to develop alkali-rich transition-metal oxides with enhanced properties for energy storage applications.

## Methods

**Synthesis**. The Mn-rich ($Li_xNi_{0.2}Mn_{0.6}O_y$, $0.00 \leq x \leq 1.52$) oxides were prepared by the following synthesis process,

$$\underbrace{0.2Ni^{2+} + 0.6Mn^{2+} + 2OH^- \xrightarrow{NH_3 \cdot H_2O} \xrightarrow{air-drying} precursor}_{hydroxide\ co-precipitation\ reaction}$$

$$\underbrace{precursor + \frac{x}{2}Li_2CO_3 \xrightarrow{850°C, 12h, air} Li_xNi_{0.2}Mn_{0.6}O_y}_{high-temperature\ lithiation\ reaction}$$

$$(0.00 \leq x \leq 1.52)$$

Firstly, a 2 M aqueous solution of stoichiometrically mixed $NiSO_4 \cdot 6H_2O$ and $MnSO_4 \cdot H_2O$ was pumped into a reactor at the adding rate of 2 ml min$^{-1}$. Simutaneously, 4 M NaOH solution and a proper amount of $NH_3 \cdot H_2O$ solution was added into the reactor. The whole process was performed under $N_2$ atmosphere. The resultant co-precipitated particles were filtered, washed with distilled water for several times to remove the imprities like $Na^+$ and $SO_4^{2-}$ ions and dried at 100 °C for 12 h. The obtained precursor powder was thoroughly mixed with different amounts of $Li_2CO_3$ ($Li:(0.2Ni + 0.6Mn) = 0.00, 0.08, 0.24, \ldots, 1.52$) by a dry-grind and a wet-milling ethanol process. Finally, the mixture was heated preheated at 550 °C for 6 h and subsequently calcined at 850 °C for 12 h in a conventional furnace with air atmosphere to obtain the materials with various lithium concentrations. For comparison with $Li_{1.2}Ni_{0.2}Mn_{0.6}O_2$, the samples are correspondingly labeled as L0.00, L0.08, L0.24, …, L1.52, respectively.

**Synchrotron radiation diffraction**. Ex situ SRD data of samples (i.e., L0.00-L1.52 and the electrodes) and in situ SRD results of coin cell (CR2025) with L0.40, L0.88 and L1.28 electrode were collected at the high-resolution Materials Science and Powder Diffraction (MSPD) beamline at ALBA, Spain, using synchrotron radiation with an energy of 30 keV. The high-temperature SRD experiments were performed at beamline P02.1, storage ring PETRA-III at DESY (Deutsches Elektronensynchrotron) in Hamburg, Germany, using synchrotron radiation with an energy of 60 keV. The prepared L0.00 was mixed with a desired amount of $Li_2CO_3$. The quartz capillary with the mixture was heated in a ceramic oven, in air, from room temperature to 850 °C by using the X-ray image processing program Fit2D. The ex situ powder diffraction experiments of selected eletrodes were also performed at beamline P02.1. All the cycled materials were measured in the discharged state.

**Neutron powder diffraction**. The NPD measurements were performed at the high-resolution powder diffractometer SPODI, research neutron reactor MLZ/FRM II in Munich, Germany, at ambient temperature. The powder was filled into a cylindrically thin-wall vanadium container with diameter of 10 mm.

**Pair distribution function analysis**. PDF analysis was performed at room temperature using the instrument at beamline P02.1 at DESY, Germany.

**X-ray absorption spectroscopy**. Ex situ X-ray absorption spectroscopy (XAS) measurements were performed at XAS beamline of synchrotron radiation source at KIT, and beamline P64 at PETRA-III, Germany. In situ XAS data of a L1.28/Li coin cell was obtained at the CLAESS beamline at ALBA synchrotron Light facility (Barcelona, Spain).

**Nuclear magnetic resonance spectroscopy**. Solid-state $^7Li$ MAS-NMR experiments were performed on a Bruker Avance 200 MHz spectrometer equipped with a 1.3 mm MAS probe at a spinning speed of around 60 kHz and a magnetic field of 4.7 T.

**Scanning electron microscopy**. The morphology of the materials was detected by a Zeiss Merlin scanning electron microscope (SEM) using an acceleration voltage of 10 kV.

**Elemental analysis**. The concentration of chemical compounds Li, Ni, Mn was determined by Inductively coupled plasma optical emission spectroscopy (ICP-OES, OPTIMA 4300 DV, PerkinElmer). Oxygen was determined by carrier gas hot extraction, which is known as the insert gas fusion method (LECO TC 600).

**Bragg coherent diffraction imaging**. BCDI measurements were performed at the ID01 beamline, ESRF—The European Synchrotron. An 8 keV coherent focused X-ray beam (500 × 500 nm FWHM) was used to illuminate a single crystallite from the from the cathode material extracted post mortem from the L1.28 coin cell after long term cycling.

**Battery tests**. The electrochemical performance of the prepared electrode materials was investigated by galvanostatic cycling in a coin-type half-cell (CR2032). The coin cells were assembled in an argon filled glove box (MBraun) with lithium metal (15.6 mm diameter, 250 μm thickness) as anode, 200 μL LP30 electrolyte (BASF) and two layers of Celgard 2325 membrane as separator. The half-cell measurements were conducted at a current density of C/10 (1C = 320 mA g$^{-1}$) at 25 °C using a VMP3 multi-channel potentiostat (Bio-Logic, France).

Further details on the synthesis procedure and the characterization techniques can be found in in the Supplementary Information.

## Data availability

The data that support the findings of this study are available from the corresponding author upon request.

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

## Acknowledgements

This work was supported by the Deutsche Forschungsgemeinschaft (DFG) under SFB 595 "Electrical Fatigue in Functional Materials" (project T3), the Bundesministerium für Bildung und Forschung (BMBF) supports Energy Research With Neutrons (ErwiN) with grant no. 05K16VK2. W.H. received financial support from the China Scholarship Council (CSC) and the Helmholtz—OCPC Postdoc-Program. We gratefully acknowledge beamline ID01 at ESRF— The European Synchrotron for the provision of synchrotron radiation facilities for the BCDI measurements. Part of these experiments were performed at MSPD beamline and CLAESS beamline at ALBA Synchrotron with the collaboration of ALBA staff and CALIPSOplus funding (Grant 730872). We gratefully acknowledge Alexander Schökel and Martin Etter at beamline P02.1 at PETRA-III for the synchrotron-based diffraction experiments, Stefan Mangold at KIT synchrotron, Vadim Murzin and Akhil Tayal at beamline P64 at PETRA-III for the XAS measurements, Udo Geckle for the SEM experiments. Additionally, we thank Shulei Chou at University of Wollongong for constructive suggestions. This work contributes to the research performed at CELEST (Center for Electrochemical Energy Storage Ulm-Karlsruhe).

## Author contributions

W.H. conceived the idea, designed the experiments, and discussed with B.S., S.W., M.K., S.J.L., A.S., C.R., M.Y., J.B., S.I., C.P.G., and H.E.; W.H. and S.W. carried out the preparation experiments. The data was measured and analyzed by W.H., B.S., M.K., S.J.L., A.S., C.R., M.Y., and S.I. All authors contributed to the writing or revision of the manuscript, and have given the approval to the final version of the manuscript.

## Competing interests

The authors declare no competing interests.
