## [Peer Review File · Nature Communications]

Reviewers' comments:

Reviewer #1 (Remarks to the Author):

This manuscript probes the formation, evolution and voltage degradation of high-energy Li- and Mn-rich layered oxides. The authors use various analytical tools to explore a grand potential phase space of $\text{Li}_x(\text{Ni}_{0.2}\text{Mn}_{0.6})\text{O}_y$, where Li content(x) is varied by changing the amount of Li_2CO_3 precursor and oxygen(y) in the final product is an open element. The authors use this information to explain the electrochemical properties (voltage fading) of $\text{Li}_{1.2}\text{Ni}_{0.2}\text{Mn}_{0.6}\text{O}_2$. This work is very solid in terms of structure analysis, and there are not many to disagree about their analysis. Unfortunately, the majority of the claims in this paper are known in the field, and there are unsupported or irrelevant claims about their use of data. Therefore, this reviewer believes that this paper is not suited for publication in Nature Communications.

(Comments)

1. How are Li- and Mn-rich layered oxides formed during synthesis?

This reviewer finds that this part (especially the title) is written a bit misleadingly although data itself can be useful. Here, the authors used terms such as "growth" or "evolution" and tried to relate their results to interpret how Li-rich and Mn-rich materials form during synthesis. Nevertheless, essentially what the authors showed in this section is a portion of a Li-Ni-Mn-O (Grand potential) phase diagram. The appearance of different phases as a result of a varied Li-precursor amount (or any other factors) is thermodynamically dictated, and strictly speaking there is no need to bring any "kinetics" when seeing a thermodynamic phase diagram. So, terms like "growth" or "evolution" "during" synthesis are confusing. In fact, this reviewer rather expected to see data like in Fig. 4 when someone says about the "formation of a compound during synthesis". Thermodynamic stability information is, of course, valuable, but this reviewer finds it is not relevant in this context in explaining "how the Li-rich cathodes form during synthesis". This reviewer would not care to know what the intermediate phases are when Li content is small at a high temperature synthesis, if Li-rich cathodes were to be made because one would immediately use sufficient amount of Li precursor in the beginning. Also, many of their conclusions are already known or predictive: LMLO phase stability as a function of Li content and O content (O chemical potential), spinel-rocksalt appearance at a low Li content, etc.

2. Monoclinic layered $\text{Li}[\text{Li}_{0.2}\text{Ni}_{0.2}\text{Mn}_{0.6}]\text{O}_2$ cathode with ultra-long cycling life

First of all, the authors claimed that their findings provided the prerequisites for the synthesis of lithium insertion compounds with high performance. How would the provided data from the section "How are Li- and Mn-rich ..." help to control the synthesis process precisely? Basically, it is well known in the Li-ion/Li-rich cathode field that one needs a certain amount of excess Li precursors when synthesizing Li-containing cathodes to account for Li evaporation at a high temperature. Also, the information in the previous section only contains a limited phase diagram along the Li-content axis and fixed Ni-Mn ratio, and also the temperature and other conditions (e.g., partial oxygen pressure in the furnace) are fixed. So, how do the findings help to precisely control the synthesis conditions to make "ultra-long cycling life" Li-rich cathodes? Also, making a pure phase does not translate into making ultra-long-cycle-life and high-performance Li-rich cathodes, as they have intrinsic limitations to address. The authors' claims can be justified only when making a pure phase has been the limiting factor for achieving high-performance cathodes, which obviously is not.

Also, this reviewer does not understand how the authors can argue "ultra-long cycling life" when the capacity and voltage fading is so obvious during their lifetime (Fig. 3). Cycling for a long time is not long-cycle-life for a cathode. If the authors want to argue that their material is indeed a long cycle life material, they at least need to show a comparison of data which shows poorer performance for the poor selection of synthesis conditions.

Finally, the authors claimed "reversible oxygen redox" during the initial ~300 cycles, and claim that further cycles involve spinel/rocksalt phase formation. There is a fast capacity fading from ~240 mAh/g to ~210 mAh/g during the initial 50 cycles. Also, the voltage fading is already very

severe after 50 cycles. Therefore, it appears very confusing to say “reversible oxygen redox during early cycling” when there are such obvious changes in the performance, which should be due to the formation of spinel/rocksalt domains already. In addition, the capacity loss can come from many factors. This reviewer suspects that the capacity decay past 200 cycles is due to the accumulation of HF in the electrolyte during extended high voltage cycling and corrosion of the cathode, which is well known to occur in the Li-ion battery field, not due to the sudden appearance of spinel/rocksalt-phases “after 200 cycles”. If the authors want to prove that their analysis is correct, they need to characterize their material upon (i) early, (ii) mid, and (iii) later cycling and show that during the early stage of cycling they do not observe spinel or rocksalts.

3. Why and how do Li-rich layered cathode materials degrade?

The degradation mechanism of the Li-rich cathodes described in this section (accumulation of spinel and rocksalt domains) has been well understood in the field. For example, the same observations and mechanisms were described in 2015 by Jainming Zheng et al. (Chem. Mater. 2015, 27, 4, 1381-1390).

4. Origin of oxygen incorporation during synthesis of Li-rich oxides

This reviewer finds that this part could have been much more interesting if the authors used this knowledge to actually suggest ways to make better Li-rich cathodes. At this point, all the information is too much of a physical interpretation of their data without proposing any guidelines. The authors should provide more discussions: (1) Why did the author compare the multi-step calcination and fast treatment process? (2) How does this change in the treatment (or precursor-type) affect the final structure, and therefore influence the performance? This reviewer feels unfortunate that the authors did very careful and detailed experiments, but I really cannot find new insights written that would help to make better Li-rich cathodes based on this manuscript.

Reviewer #2 (Remarks to the Author):

This manuscript reports very interesting results using well-designed in situ characterizations on synthesis/heating experiments to understand the thermodynamic driving force of the degradation process of Li- and Mn-rich layered oxides cathode materials. The conclusions are well supported by the experimental results and the data interpretation. The layered (ordered) rock salt to spinel transition/degradation has been long known in the field of Li-ion batteries. However, this process was not fully revealed in great details before this work. Using careful in situ characterizations to investigate the reactions between transition metal oxide and Li_2CO_3 is a unique approach to understand the thermodynamics and kinetics of layered oxides with respect to the Li and O contents. The publication of this work can bring new insights of the layered oxide cathodes to the field and inspire the materials design and battery performance improvement. I'd suggest this work to be published in Nature Communications. Some minor questions and comments are provided below. But they are optional and will not change my recommendation.

Figure 3. It should be labeled in the figure or stated in the caption the cycle number of the “cycled electrode”

What is the shoulder on the left side of the 131 peak in

Figure 3 around 11.6 degree 2-theta?

Figure 4. Since the most important phase transformations/evolutions take place during the heating in the low temperature range in (a) (e.g. 0-100 minutes). It will be helpful to provide a zoomed-in image of the low temperature range in the supplementary information that can better present the spinel to disordered rocksalt transition.

It's a bit confusing in line 417-420 “For the fast thermal treatment process, the Li-free spinel phase rapidly converts to the intermediate phases, i.e. layered Li_2TMO_2 phase (P3m1), spinel phases (Fd3m and I41/amd) and rock-salt phase (Fm3m), in the early stage of the reaction, see

Figure 4(c & d) and Table S4" The P3m1 phase was not labeled in Figure 4. Was it included in the phase fraction calculation or not?

Line 463-464 "i.e. a kind of partially reversible formation of Li-rich phase (lithium extraction accompanying oxygen evolution), see Figure 5." This statement is probably right in terms of thermodynamics. But it might be worth it to note that the kinetics of the degradation reaction, particularly the rate-limiting steps may be different from what is observed in the synthesis reactions. I.e. the synthesis seems to be a surface-bulk-limited reaction, while it is not clear what the kinetics of the degradation reaction would be like.

Reviewer #3 (Remarks to the Author):

The manuscript discussed: i) structure evolution of Li-Ni-Mn-O system with changing Li composition from the high temperature synthesis. ii) structure evolution of layered $\text{Li}_{1.2}\text{Ni}_{0.2}\text{Mn}_{0.6}\text{O}_2$ after long electrochemical cycling. iii) structure evolution of Li-free spinel plus Li precursor with increasing synthesis temperature up to high temperatures. iv) Some understandings by connecting the above points i), ii), iii).

The results are in general interesting. Points i) and iii) (Fig. 1 and Fig. 4) are straightforward and solid. However, point ii) (Fig. 2, 3) on the electrochemical structure evolution is not convincing. I am also skeptical about the methodology that the authors tried to make the connections among i) ii) iii). Thus I cannot recommend the manuscript to be published on Nature Communications.

Although it is nice to show the long cycling battery test (Fig. 2c), it is not clear how representative it is, i.e. do they always drop the capacity quickly in the first ~50 cycles, then stabilize to ~300 cycles, beyond which drop more quickly following a fixed slope? I hope that the authors had a few batteries running simultaneously at the same or different rates. If such trend is general, it will be valuable to show additional ex situ characterizations after each stage, i.e., after 50 cycles, 300 cycles, etc. That will help understand why the capacity plateau ends at around 300 cycles, as I feel it lacks an articulation about why the gradual oxygen loss beyond a certain point (such as beyond 300 cycles) can lead to an abrupt slope change in Fig. 2c. What is the critical structural change that ends the capacity plateau at ~300 cycles?

Furthermore, although it is interesting to make the comparison between phase evolution at high temperature synthesis conditions and that at room temperature electrochemical cycling conditions, I am not sure the authors made it clear the fundamental connection in the very different temperature scales. At high temperature, many kinetic barriers can be easily overcome, such as the oxygen vacancy diffusion barrier or transition metal migration barrier, while at room temperature, these barriers could largely limit or even forbid a thermodynamic phase transformation. Thus, it is not clear why the high temperature phase evolution can give insights to the room temperature one here. In my opinion, it will be more valuable to focus on the differences between different temperature scales, and discuss more about the unique electrochemical structural evolution at room temperature.

Point-by-point response (in blue) to the reviewers' comments

We thank the reviewers for their constructive ideas, which have helped us to substantially improve the quality of our manuscript. The revised manuscript includes a series of new *ex situ* synchrotron radiation diffraction and X-ray absorption spectroscopy results of the Li-rich electrode after different cycle numbers and state-of-the-art Bragg coherent diffraction imaging (BCDI) of a single particle obtained after one year of cycling. Below we list detailed responses to the original reviewers' comments.

Reviewer #1:

This manuscript probes the formation, evolution and voltage degradation of high-energy Li- and Mn-rich layered oxides. The authors use various analytical tools to explore a grand potential phase space of $\text{Li}_x(\text{Ni}_{0.2}\text{Mn}_{0.6})\text{O}_y$, where Li content(x) is varied by changing the amount of Li_2CO_3 precursor and oxygen(y) in the final product is an open element. The authors use this information to explain the electrochemical properties (voltage fading) of $\text{Li}_{1.2}\text{Ni}_{0.2}\text{Mn}_{0.6}\text{O}_2$. This work is very solid in terms of structure analysis, and there are not many to disagree about their analysis. Unfortunately, the majority of the claims in this paper are known in the field, and there are unsupported or irrelevant claims about their use of data. Therefore, this reviewer believes that this paper is not suited for publication in Nature Communications.

We thank the reviewer for her/his assessment and suggestions. Her/his constructive remarks helped us improve the clarity of our manuscript, as detailed below.

Comment #1: How are Li- and Mn-rich layered oxides formed during synthesis? This reviewer finds that this part (especially the title) is written a bit misleadingly although data itself can be useful. Here, the authors used terms such as "growth" or "evolution" and tried to relate their results to interpret how Li-rich and Mn-rich materials form

during synthesis. Nevertheless, essentially what the authors showed in this section is a portion of a Li-Ni-Mn-O (Grand potential) phase diagram. The appearance of different phases as a result of a varied Li-precursor amount (or any other factors) is thermodynamically dictated, and strictly speaking there is no need to bring any “kinetics” when seeing a thermodynamic phase diagram. So, terms like "growth" or "evolution" "during" synthesis are confusing. In fact, this reviewer rather expected to see data like in Fig. 4 when someone says about the “formation of a compound during synthesis”. Thermodynamic stability information is, of course, valuable, but this reviewer finds it is not relevant in this context in explaining “how the Li-rich cathodes form during synthesis”. This reviewer would not care to know what the intermediate phases are when Li content is small at a high temperature synthesis, if Li-rich cathodes were to be made because one would immediately use sufficient amount of Li precursor in the beginning. Also, many of their conclusions are already known or predictive: LMLO phase stability as a function of Li content and O content (O chemical potential), spinel-rocksalt appearance at a low Li content, etc.

Answer #1: Thank you very much for your comments. We agree that this section provides a portion of a Li-Ni-Mn-O (Grand potential) phase diagram. This thermodynamic phase diagram reveals a tendency of Mn-rich oxides towards formation of different phases, depending on the Li content in the system, during synthesis and electrochemical cycling of Li- and Mn-rich layered cathode materials. All the terms such as "growth" or "evolution" have been modified in our updated manuscript.

We do not fully agree with the reviewer’s statement that it is not relevant in this context in explaining how the Li-rich cathodes form during synthesis, because for the formation of the final products the system has to overcome the activation energy of the intermediate phases although it is a thermodynamically dictated reaction. The structural and electronic changes of thermally stable $\text{Li}_x\text{Ni}_{0.2}\text{Mn}_{0.6}\text{O}_y$ oxides as a function of Li and O content (**Figure 1**) are in excellent agreement with the structural evolution of Li-free spinel phase with incorporation of Li and O during synthesis,

please see **Figure 5(a&b)** in the new manuscript.

We also do not agree with the reviewers' statement that many of our conclusions are already known or predictive. The appearance of spinel/rock-salt phases at a low Li content was already reported in the literature.¹ However, the kinetics of spinel/rock-salt-to-layered transition is still poorly understood, especially under nonequilibrium conditions, please see the reaction pathways for the formation of Li-rich materials in **Figure 5 & 6**.

Additionally, several other fundamental questions have been addressed in this section. Please see below,

1. Both L0.00 and L0.40 in **Figure 1** have a spinel AB_2O_4 structure ($Fd-3m$) with the same content of Ni and Mn, while the tetrahedral and octahedral positions in their spinel matrix are almost totally occupied, so how and where are lithium/oxygen incorporated into spinel architecture? How in turn does Li/O incorporation influence the microstructure and local crystallography of spinel oxides?
2. Generally, the Mn^{3+} state can cause a tetragonal distortion in pure Mn spinel structure (Mn_3O_4) due to the Jahn-Teller effect.^{2,3} Why does L0.00 with the dominantly Mn^{3+} state still possess a cubic spinel phase? Which elements are located at tetrahedral sites in the Li-free spinel oxide (L0.00)? Only Mn or both Ni and Mn?
3. Due to the fact that spinel/rock-salt-type/layered phase have the common cubic-close packed (ccp) oxygen framework, how is oxygen incorporated into or released from the ccp oxygen lattice in the crystal structure during formation (Li insertion) and degradation (Li extraction) of Li-rich layered oxides?
4. As we know, the Li-rich layered materials can be made by a mixture of Li source and Li-free precursor during thermal treatment, but how many Li/O species are exactly incorporated into the host crystal structure to form the

expected final products during synthesis?

5. If a huge amount of Li precursor is provided in the beginning, what happens structurally and electronically to final layered materials (i.e. $1.28 \leq x \leq 1.52$ in $\text{Li}_x\text{Ni}_{0.2}\text{Mn}_{0.6}\text{O}_y$)? Do these obtained 'Li-rich' layered oxides still have the same electrochemical performances as L1.28 electrode?

Furthermore, we map the three-dimensional (3D) strain field inside a single crystallite of the cycled L1.28 cathode (after 868 cycles) with Bragg coherent diffraction imaging (BCDI), for the first time, to better understand the kinetics of the degradation reaction, please see **Figure 4** in our new manuscript (page 18).

Therefore, consistent results obtained from thermodynamic and kinetic study of $\text{Li}_x\text{Ni}_{0.2}\text{Mn}_{0.6}\text{O}_y$ oxides unambiguously confirm that the formation, evolution and degradation of Li-rich layered cathode materials are related as supported by the data.

Comment #2: Monoclinic layered $\text{Li}[\text{Li}_{0.2}\text{Ni}_{0.2}\text{Mn}_{0.6}]\text{O}_2$ cathode with ultra-long cycling life First of all, the authors claimed that their findings provided the prerequisites for the synthesis of lithium insertion compounds with high performance. How would the provided data from the section "How are Li- and Mn-rich ..." help to control the synthesis process precisely? Basically, it is well known in the Li-ion/Li-rich cathode field that one needs a certain amount of excess Li precursors when synthesizing Li-containing cathodes to account for Li evaporation at a high temperature. Also, the information in the previous section only contains a limited phase diagram along the Li-content axis and fixed Ni-Mn ratio, and also the temperature and other conditions (e.g., partial oxygen pressure in the furnace) are fixed. So, how do the findings help to precisely control the synthesis conditions to make "ultra-long cycling life" Li-rich cathodes? Also, making a pure phase does not translate into making ultra-long-cycle-life and high-performance Li-rich cathodes, as they have intrinsic limitations to address. The authors' claims can be justified only when making a pure phase has been the limiting factor for achieving high-

performance cathodes, which obviously is not. Also, this reviewer does not understand how the authors can argue “ultra-long cycling life” when the capacity and voltage fading is so obvious during their lifetime (Fig. 3). Cycling for a long time is not long-cycle-life for a cathode. If the authors want to argue that their material is indeed a long cycle life material, they at least need to show a comparison of data which shows poorer performance for the poor selection of synthesis conditions. Finally, the authors claimed “reversible oxygen redox” during the initial ~300 cycles, and claim that further cycles involve spinel/rocksalt phase formation. There is a fast capacity fading from ~240 mAh/g to ~210 mAh/g during the initial 50 cycles. Also, the voltage fading is already very severe after 50 cycles. Therefore, it appears very confusing to say “reversible oxygen redox during early cycling” when there are such obvious changes in the performance, which should be due to the formation of spinel/rocksalt domains already. In addition, the capacity loss can come from many factors. This reviewer suspects that the capacity decay past 200 cycles is due to the accumulation of HF in the electrolyte during extended high voltage cycling and corrosion of the cathode, which is well known to occur in the Li-ion battery field, not due to the sudden appearance of spinel/rocksalt-phases “after 200 cycles”. If the authors want to prove that their analysis is correct, they need to characterize their material upon (i) early, (ii) mid, and (iii) later cycling and show that during the early stage of cycling they do not observe spinel or rocksalts.

Answer #2: We greatly appreciate the reviewers’ detailed and constructive comments. The goal of fixing other synthetic conditions such as temperature, Ni-Mn ratio and partial oxygen pressure is to investigate the effects of Li/O content on the structural and electronic properties of Li-containing oxides in detail. The findings in the previous section provide a precise mapping for the synthesis of lithium insertion compounds with superior performance, i.e. the precise preparation of monoclinic layered $\text{Li}[\text{Li}_{0.2}\text{Ni}_{0.2}\text{Mn}_{0.6}]\text{O}_2$ oxide. A comparison of data, which shows worse performance for the poor selection of synthesis conditions, has been added in our updated supporting information, see **Figure S7**.

Ex situ XRD/XAS results after 50 and 200 cycles have been included in our new manuscript. Very importantly, we only observe the formation of partially disordered Li-containing rock-salt phase ($Fm\bar{3}m$) during the early stage of cycling (200 cycles). Thermodynamically driven formation of the Li-poor spinel phase (AB_2O_4 , $Fd\bar{3}m$) is finally observed after ultra-long cycling time (868 cycles). Such long-term cycling battery test could give valuable information on the kinetic behavior of phase transition from a Li-rich layered $Li[Li_xTM_{1-x}]O_2$ ($C2/m$ and/or $R\bar{3}m$) to a Li-containing rock-salt $[Li_xTM_{1-x}]O$ ($Fm\bar{3}m$) and then to a Li-poor spinel $(Li_xTM_{1-x})[TM_2]O_4$ ($Fd\bar{3}m$) structure. The good durability (within the first 300 cycles) provides a clear evidence that the oxygen redox process is at least partially reversible within a relatively long period, contributing to a specific capacity of $\sim 100 \text{ mA h g}^{-1}$, because the Ni^{2+}/Ni^{4+} redox activity can only compensate for 0.4 Li ion extraction from $Li_{1.2}Ni_{0.2}Mn_{0.6}O_2$ corresponding to the remaining capacity of $\sim 100 \text{ mA h g}^{-1}$.

The related discussion has been added to our revised manuscript, please also see below.

Main text:

After 50 and 200 cycles, the reflections in the SRD patterns of the cycled L1.28 electrodes can be indexed to a layered structure with the space symmetry of $C2/m$ and/or $R\bar{3}m$, see **Figure S9**. Compared to the SRD pattern of L1.28 electrode before cycling, all reflections in the SRD pattern of L1.28 cathode after 200 cycles move to lower 2-theta angles (lattice expansion), the split reflections of $-133_m/33-1_m$ or $018_h/110_h$ tend to merge into a single reflection (less distortion from a cubic phase). The oxidation state of TM in the cycled electrode does not change substantially and can still be assigned to Ni^{2+} and Mn^{4+} state (**Figure S9(c)**). Thus, it would be reasonable to speculate that the expansion of the Li-rich layered unit cell is caused by the formation of more disordered $[Li/TM]O_6$ octahedra, i.e. from Li-O-[Li/TM] chains in the ordered layered structure to partially $[Li/TM]-O-[Li/TM]$ linkages in the disordered rock-salt structure, which is closely related to the lattice contraction

behaviour in the fully disordered Li-containing rock-salt structure during synthesis of LMLOs in **Figure 5**. Additionally, the TM occupancy on the Li sites in the layered structure of L1.28 electrode has increased from $\sim 3(2)$ % before cycling to $\sim 6(2)$ % after 200 cycles (Rietveld refinement results in **Figure S9** and **Table S4**), again suggesting the generation of partially disordered Li-containing rock-salt-type phase ($Fm\bar{3}m$). On the other hand, lack of evidence for the spinel phase (AB_2O_4 , $Fd\bar{3}m$) formation at this stage indicates that the cycled L1.28 electrode does not reach equilibrium within five months cycling after a small amount of irreversible Li/O loss (see **Figure 1**). Therefore, the commonly called ‘spinel-like’ phase formed within a limited period can be interpreted as the accumulation of layered ($C2/m$ or $R\bar{3}m$) & Li-containing rock-salt $[Li_xTM_{1-x}]O$ ($Fm\bar{3}m$) nano-domains in our electrodes, which agrees quite well with the analysis of activation process (layered-to-rock-salt transition) of L1.20 electrode during the first 100 cycles in **Figure S7 & S8**. The disordered rock-salt-type phase can block the lithium diffusion pathway to some extent and thus result in a worsening electrochemical performance.⁴ (page 14 - 15)

Taken together, thermodynamically driven formation of the Li-poor spinel phase (AB_2O_4 , $Fd\bar{3}m$) is finally observed after ultra-long cycling time, which arises from irreversible release of lithium and oxygen from layered crystal lattice and is supposed to be located at the surface (see **Figure 4**). Unfortunately, the accumulation of layered & rock-salt-type & spinel domains with the large lattice mismatch induces a much higher diffusion barrier for Li ions, compared to the stage II, and thereby a considerably capacity loss at stage III in **Figure 2(c)**. In addition, the capacity fading is probably also a consequence of the slow dissolution of manganese (III) ions in the spinel phase ($2Mn^{3+} \rightarrow Mn^{4+} + Mn^{2+}$), the lithium dendrite and SEI growth in the coin cell, and electrolyte evaporation.⁵⁻⁸ Most importantly, such long-term cycling battery test could give valuable information on the kinetic behavior of phase transitions from a Li-rich layered $Li[Li_xTM_{1-x}]O_2$ ($C2/m$ and/or $R\bar{3}m$) to a partially disordered Li-containing rock-salt $[Li_xTM_{1-x}]O$ ($Fm\bar{3}m$) and then to a Li-poor spinel $(Li_xTM_{1-x})[TM_2]O_4$ ($Fd\bar{3}m$) structure. (Page 17)

Supplementary information:

Figure S7. Cycling properties of (a) L1.04 electrode with spinel & rock-salt-type & layered phases, (c) L1.20 electrode with mainly layered rock-salt & Li-containing phases, (e) L1.28 electrode and (g) L1.52 electrode with monoclinic layered phase; the corresponding charge–discharge voltage profiles of the selected electrodes (b, d, f, h) between 2.0 and 4.8 V at a current density of 32 mA g^{-1} at room temperature.

Figure S7 shows the electrochemical performance and the selected charge–discharge voltage profile of four typical samples, i.e. L1.04, L1.20, L1.28 and L1.52. When compared with the layered L1.28 and L1.52 cathodes, both L1.04 and L1.20 electrodes with multiple layered/spinel/rock-salt-type phases exhibit a poor cycling

performance and a severe voltage decay upon cycling. Regarding the L1.20 electrode, after 100 cycles at charge/discharge rate of 0.1 C, nearly 500 % of the initial discharge capacity can be achieved, indicating a continuous electrochemically induced activation process of Li-containing/defective disordered rock-salt phase during cycling, see **Figure S8**. Note that the L1.20 electrode experiences a rapid capacity loss after activation process (i.e. 100 cycles), which probably results from the decomposition of active materials, mostly Mn^{3+} in the spinel phase (see the plateau at ~ 2.7 V), after the formation of Li-poor spinel phase caused by successive loss of Li/O. Obviously, the L1.52 cathode also suffers from a serious voltage fading during cycling, but the plateau of L1.52 cathode at about 2.7 V corresponding to the characteristics of spinel phase is much smaller than that of L0.88 and L1.20 electrodes, suggesting that the voltage degradation of Li-rich layered cathodes is closely tied to the formation of Li-containing/defective rock-salt phase. Keep in mind that the oxidation state of nickel in the L1.52 electrode is higher than 2+, see **Figure 1(c)**, its capacity, thereby, mostly originates from oxygen redox during cycling. The oxygen anions can be easily removed from layered structure of L1.52 electrode when a large number of electrons are released from oxygen $2p$ orbitals, thus resulting in a severe phase transition from layered to rock-salt/spinel structure, and consequently a poor electrochemical performance. Therefore, the electrochemical properties are strongly related to the phase and chemical composition of the cathode materials.

Figure S8. Synchrotron radiation diffraction patterns of (a) L1.20 electrode before cycling and after 100 cycles; Rietveld refinement against synchrotron radiation diffraction patterns of L1.20 electrode (b) before cycling and (c) after 100 cycles. These results reveal that the phase transformation from ordered layered ($C2/m$) to defective/Li-containing disordered rock-salt ($Fm\bar{3}m$) structure is responsible for the activation process of L1.20 electrode, rather than the formation of spinel ($Fd\bar{3}m$) phase during the first 100 cycles. Note that in order to show the compositional changes of layered, rock-salt and spinel phases after activation process, the phase fraction of Li_2CO_3 in the L1.20 electrode after 100 cycles was not accounted for in Figure (c).

Figure S9. Synchrotron radiation diffraction patterns of (a) L1.28 electrode before cycling and after 50, 200 cycles; Rietveld refinement against synchrotron radiation diffraction patterns of L1.20 electrode (b) after 100 cycles; (c) Ni K-edge and Mn K-edge XAS spectra of the L1.28 electrode before cycling and after 200 cycles.

Comment #3: Why and how do Li-rich layered cathode materials degrade? The degradation mechanism of the Li-rich cathodes described in this section (accumulation of spinel and rocksalt domains) has been well understood in the field. For example, the same observations and mechanisms were described in 2015 by Jianming Zheng et al. (Chem. Mater. 2015, 27, 4, 1381-1390).

Answer #3: We thank the reviewer for the comments and recommending the paper by Jianming Zheng et al (2015) (Title: Structural and Chemical Evolution of Li- and Mn-Rich Layered Cathode Material), although we respectfully disagree with some of the points in the reference, as discussed below.

A possible mechanism, i.e. the $\text{Li}[\text{Li}_{0.2}\text{Ni}_{0.2}\text{Mn}_{0.6}]\text{O}_2$ cathode experiencing a phase transformation from the layered structure (initial $C2/m$ phase transforms to $R-3m$ phase after activation) to a LT-LiCoO₂ type defect spinel-like structure (with the $Fd-3m$ space group) and then to a disordered rock-salt structure (with the $Fm-3m$ space group), was proposed in the previous research¹. However, this is contrary to thermodynamic and dynamic aspects of the degradation of Li-rich layered materials, see **Figure 1 & 5** in our new manuscript. The fundamental questions in the field of Li-ion batteries that need to be clarified are (1) what is the nature of this so-called spinel-like ($Fd\bar{3}m$) phase? and (2) whether there is a rate-limiting step before the formation of spinel, AB_2O_4 , phase ($Fd\bar{3}m$) during cycling. Such a reaction would involve a huge amount of Li/O release during the phase transition from Li-rich layered ($\text{Li}_{1.2}\text{Ni}_{0.2}\text{Mn}_{0.6}\text{O}_2$) to Li-free rock-salt (TMO, $Fm\bar{3}m$) structure.

In the reference¹, the authors found that the intensity plot along the Li layer displays a low intensity in one octahedral site followed by a high intensity in the adjacent octahedral site by using high-resolution STEM technique. Thus, the LT-LiCoO₂ type defect spinel-like structure model was proposed based on the STEM images from [011] zone axis. As a kind reminder, TEM images are projection images. Since the transformation from cubic (rock-salt) to rhombohedral (layered) structure exhibits four degrees of freedom concerning the rhombohedral axis (stacking direction), how to exclude the possibility of an overlapping nano-domain of Li-rich layered phase with two different orientations and Li-containing cubic rock-salt phase?

Our long-term cycling battery test and BCDI results provide valuable information on the kinetic behavior of phase transition from a Li-rich layered $\text{Li}[\text{Li}_x\text{TM}_{1-x}]\text{O}_2$ ($C2/m$ and/or $R\bar{3}m$) to a partially disordered Li-containing rock-salt $[\text{Li}_x\text{TM}_{1-x}]\text{O}$ ($Fm\bar{3}m$) and then to a Li-poor spinel $(\text{Li}_x\text{TM}_{1-x})[\text{TM}_2]\text{O}_4$ ($Fd\bar{3}m$) structure, which are different with the proposed mechanisms in the literature¹.

Comment #4: Origin of oxygen incorporation during synthesis of Li-rich oxides This

reviewer finds that this part could have been much more interesting if the authors used this knowledge to actually suggest ways to make better Li-rich cathodes. At this point, all the information is too much of a physical interpretation of their data without proposing any guidelines. The authors should provide more discussions: (1) Why did the author compare the multi-step calcination and fast treatment process? (2) How does this change in the treatment (or precursor-type) affect the final structure, and therefore influence the performance? This reviewer feels unfortunate that the authors did very careful and detailed experiments, but I really cannot find new insights written that would help to make better Li-rich cathodes based on this manuscript.

Answer #4: We are grateful to the reviewer for recognizing the value of this part in our work. The multi-step slow calcination process was firstly carried out to elucidate what happens to Li-free spinel oxide when Li/O ions are gradually incorporated into its matrix architecture. The *in situ* diffraction results reveal that high-temperature lithiation reaction is a multistep reaction, Li/O-incorporation-induced structural evolution in Li-free spinel oxide is in good agreement with the analysis in **Figure 1**.

Considering that the structural evolution of Li-rich layered oxides during electrochemical cycling is far away from thermal equilibrium (see **Figure 3**), the fast thermal treatment process was then utilized to gain new insights into the kinetics of non-equilibrium lithiation reaction. These data unambiguously demonstrate that the nonequilibrium phase transition from Li-poor spinel to Li-rich layered structure is rate-limited by the formation of an often ignored Li-rich rock-salt-type intermediate. These results will lead to new insights into the voltage degradation of Li-rich layered cathode materials. The related discussion has been added to this section in our revised manuscript (Page 19-23).

A comprehensive and nuanced analysis of the influence of treatment (microwave heating) and precursor-type (hydroxide) on the structure and the performance of Li-rich materials was reported in our previous work². The changes in the treatment or precursor-type could affect the chemical reaction pathways, and therefore the

chemical- and phase-composition of final product. However, the formation of Li-rich cathode materials during high-temperature lithiation reaction is thermodynamically favored. When a mixture of different precursors with an appropriate amount of Li source is heated at high-temperature (e.g., 850 °C) for a long time (e.g., 12 h), the mixture would always convert to the Li-rich layered oxides.

As temperature decreases, the time needed to reach equilibrium increases exponentially, this is one reason why the thermodynamically driven formation of spinel (AB_2O_4 , $Fd\bar{3}m$) phase during degradation of Li-rich oxides is so difficult to observe in the cycled electrode within a limit cycling at room temperature. The low temperature or fast heating process will take the system far away from thermal equilibrium. During synthesis of layered $Li[Li_{1.2}Ni_{0.2}Mn_{0.6}]O_2$ oxide under rapid-heating conditions, the cubic spinel $Mn[Ni_{0.75}Mn_{1.25}]O_4$ phase (L0.00) experiences a transformation from a Li-free spinel ($Fd\bar{3}m$) to a fully disordered Li-containing rock-salt ($Fm\bar{3}m$) and finally to a Li-rich layered structure ($C2/m$) with the incorporation of lithium and oxygen. As a gradual Li/O loss during long-term cycling occurs, the Li-rich layered $Li[Li_{1.2}Ni_{0.2}Mn_{0.6}]O_2$ cathode tends to transform back to the ‘Li-poor’ spinel phase ($Fd\bar{3}m$), but this transition is limited by the generation of the Li-containing rock-salt phase ($Fm\bar{3}m$). Such an inversed relationship between formation pathway and degradation process of LMLOs is not a coincidence but intrinsic due to the fact that both occur under non-equilibrium conditions driven by Li/O concentration gradient.

Overall, based on thermodynamic and kinetic considerations, a correlation between formation and degradation of Li- and Mn-rich layered oxides (LMLOs) is established. A series of new findings linked to this subject will be reported for the first time, we believe that the conclusive evidence about structural and electrical details as well as about thermodynamic phase stability of Li-rich oxides will be of interest to both the battery community and the broader readership of *Nature Communications*.

Reviewer #2:

This manuscript reports very interesting results using well-designed in situ characterizations on synthesis/heating experiments to understand the thermodynamic driving force of the degradation process of Li- and Mn-rich layered oxides cathode materials. The conclusions are well supported by the experimental results and the data interpretation. The layered (ordered) rock salt to spinel transition/degradation has been long known in the field of Li-ion batteries. However, this process was not fully revealed in great details before this work. Using careful in situ characterizations to investigate the reactions between transition metal oxide and Li_2CO_3 is a unique approach to understand the thermodynamics and kinetics of layered oxides with respect to the Li and O contents. The publication of this work can bring new insights of the layered oxide cathodes to the field and inspire the materials design and battery performance improvement. I'd suggest this work to be published in Nature Communications. Some minor questions and comments are provided below. But they are optional and will not change my recommendation.

We thank the reviewer for recognizing the value of our work and for the insightful comments and questions.

Comment #1: Figure 3. It should be labeled in the figure or stated in the caption the cycle number of the “cycled electrode” What is the shoulder on the left side of the 131 peak in Figure 3 around 11.6 degree 2-theta?

Answer #1: We thank the reviewer for this observation. The cycle number of the “cycled electrode” has been now added in the caption of Figure 3 (page 16). We cannot find the shoulder on the left side of the 131 reflection in Figure 3 in the raw materials, so it is not related to our active material.

Comment #2: Figure 4. Since the most important phase transformations/evolutions take place during the heating in the low temperature range in (a) (e.g. 0-100 minutes).

It will be helpful to provide a zoomed-in image of the low temperature range in the supplementary information that can better present the spinel to disordered rocksalt transition.

Answer #2: A zoomed-in image of the low temperature range has been now added in our new supplementary document (Figure S22 in Page 23), please also see below.

Figure S22. Time-resolved high-temperature SRD patterns of a mixture of Li_2CO_3 and Li_2CO_3 during a slow thermal treatment.

Comment #3: It's a bit confusing in line 417-420 "For the fast thermal treatment process, the Li-free spinel phase rapidly converts to the intermediate phases, i.e. layered Li_2TMO_2 phase ($P3m1$), spinel phases ($Fd3m$ and $I41/amd$) and rock-salt phase ($Fm3m$), in the early stage of the reaction, see Figure 4(c & d) and Table S4" The $P3m1$ phase was not labeled in Figure 4. Was it included in the phase fraction calculation or not?

Answer #3: The phase fraction of $P3m1$ phase was not included in the previous Figure 4, all the phase fractions have been now involved in our revised Figure 5 on Page 21.

Comment #4: Line 463-464 "i.e. a kind of partially reversible formation of Li-rich phase (lithium extraction 464 accompanying oxygen evolution), see Figure 5." This statement is probably right in terms of thermodynamics. But it might be worth it to note that the kinetics of the degradation reaction, particularly the rate-limiting steps may be different from what is observed in the synthesis reactions. I.e. the synthesis

seems to be a surface-bulk-limited reaction, while it is not clear what the kinetics of the degradation reaction would be like.

Answer #4: To better understand the kinetics of the degradation reaction, we map the three-dimensional (3D) strain ($\Delta d/d$) field inside a single crystallite of the cycled L1.28 cathode (after 868 cycles) with Bragg coherent diffraction imaging (BCDI). Details of the experimental method and data reduction can be found in the supplementary material. **Figure 4** shows a distinct homogeneous tensile strain towards the surface of the crystallite from its core, i.e. the lattice parameter is increasing. The average d_{001_m} lattice parameter of the crystal determined by BCDI from the layered structure is 0.4805(2) nm (zero-strain in **Figure 4**) and is very close to the corresponding value determined by SRD in **Figure 3c** (0.4802(2) nm). BCDI cannot distinguish unambiguously if the structure of the crystal is layered or spinel, but in this case BCDI is sensitive to both phases because the 111_s spinel would interfere with the 001_m layered reflection. However, given the XRD results from many millions of crystallites we anticipate Li-poor spinel on the very surface with a larger lattice parameter than the layered phase (L1.28, 0.4751(2) nm) and spinel $\text{LiNi}_{0.5}\text{Mn}_{1.5}\text{O}_4$ (L0.40, 0.4725(2) nm), consistent with the strain distribution observed in **Figure 4**. The Li-poor spinel layer is probably thinner than the resolution (35 nm), see **Figure S12**, of the reconstructed object image but the distortion of the lattice between the two phases extends into the crystallite, therefore is visible. Hence, the degradation mechanism of Li-rich layered cathodes can be thought of a surface-bulk-limited reaction.

Figure 4. Bragg Coherent Diffractive Imaging reveals the internal structure of a single crystallite from the L1.28 cathode after one year cycling. (a) 3D render of the crystallite, where the colour map denotes strain and corresponding cross sections through the 3D structure in the (b) XY, (c) XZ and (d) YZ planes. Tick spacings along all axes correspond to 100 nm.

Reviewer #3:

The manuscript discussed: i) structure evolution of Li-Ni-Mn-O system with changing Li composition from the high temperature synthesis. ii) structure evolution of layered $\text{Li}_{1.2}\text{Ni}_{0.2}\text{Mn}_{0.6}\text{O}_2$ after long electrochemical cycling. iii) structure evolution of Li-free spinel plus Li precursor with increasing synthesis temperature up to high temperatures. iv) Some understandings by connecting the above points i), ii), iii).

The results are in general interesting. Points i) and iii) (Fig. 1 and Fig. 4) are straightforward and solid. However, point ii) (Fig. 2, 3) on the electrochemical structure evolution is not convincing. I am also skeptical about the methodology that the authors tried to make the connections among i) ii) iii). Thus I cannot recommend the manuscript to be published on Nature Communications.

We thank the reviewer for valuable comments for improving the quality of our manuscript.

Comment #1: Although it is nice to show the long cycling battery test (Fig. 2c), it is not clear how representative it is, i.e. do they always drop the capacity quickly in the first ~50 cycles, then stabilize to ~300 cycles, beyond which drop more quickly following a fixed slope? I hope that the authors had a few batteries running simultaneously at the same or different rates. If such trend is general, it will be valuable to show additional ex situ characterizations after each stage, i.e., after 50 cycles, 300 cycles, etc. That will help understand why the capacity plateau ends at around 300 cycles, as I feel it lacks an articulation about why the gradual oxygen loss beyond a certain point (such as beyond 300 cycles) can lead to an abrupt slope change in Fig. 2c. What is the critical structural change that ends the capacity plateau at ~300 cycles?

Answer #1: We note that the sudden capacity loss of about 30 mA h g^{-1} is not always observed in the first 50 cycles (**Figure S7**), but it is the good capacity retention with

serious voltage decay (stage II) that is common in Li-rich layered cathodes within a limited cycling time (e.g. 300 cycles).^{1,9} *Ex situ* XRD/XAS results after 50 and 200 cycles have been included in our new manuscript. Very importantly, we only observe the formation of partially disordered Li-containing rock-salt phase ($Fm\bar{3}m$) during the early stage of cycling (200 cycles). Thermodynamically driven formation of the Li-poor spinel phase (AB_2O_4 , $Fd\bar{3}m$) is finally observed after ultra-long cycling time (868 cycles). Such long-term cycling battery test could give valuable information on the kinetic behavior of phase transition from a Li-rich layered $Li[Li_xTM_{1-x}]O_2$ ($C2/m$ and/or $R\bar{3}m$) to a Li-containing rock-salt $[Li_xTM_{1-x}]O$ ($Fm\bar{3}m$) and then to a Li-poor spinel $(Li_xTM_{1-x})[TM_2]O_4$ ($Fd\bar{3}m$) structure.

The related discussion has been added to our revised manuscript, please also see below.

Main text:

After 50 and 200 cycles, the reflections in the SRD patterns of the cycled L1.28 electrodes can be indexed to a layered structure with the space symmetry of $C2/m$ and/or $R\bar{3}m$, see **Figure S9**. Compared to the SRD pattern of L1.28 electrode before cycling, all reflections in the SRD pattern of L1.28 cathode after 200 cycles move to lower 2-theta angles (lattice expansion), the split reflections of $-133_m/33-1_m$ or $018_h/110_h$ tend to merge into a single reflection (less distortion from a cubic phase). The oxidation state of TM in the cycled electrode does not change substantially and can still be assigned to Ni^{2+} and Mn^{4+} state (**Figure S9(c)**). Thus, it would be reasonable to speculate that the expansion of the Li-rich layered unit cell is caused by the formation of more disordered $[Li/TM]O_6$ octahedra, i.e. from Li-O-[Li/TM] chains in the ordered layered structure to partially $[Li/TM]-O-[Li/TM]$ linkages in the disordered rock-salt structure, which is closely related to the lattice contraction behaviour in the fully disordered Li-containing rock-salt structure during synthesis of LMOs in **Figure 5**. Additionally, the TM occupancy on the Li sites in the layered structure of L1.28 electrode has increased from $\sim 3(2)$ % before cycling to $\sim 6(2)$ %

after 200 cycles (Rietveld refinement results in **Figure S9** and **Table S4**), again suggesting the generation of partially disordered Li-containing rock-salt-type phase ($Fm\bar{3}m$). On the other hand, lack of evidence for the spinel phase (AB_2O_4 , $Fd\bar{3}m$) formation at this stage indicates that the cycled L1.28 electrode does not reach equilibrium within five months cycling after a small amount of irreversible Li/O loss (see **Figure 1**). Therefore, the commonly called ‘spinel-like’ phase formed within a limited period can be interpreted as the accumulation of layered ($C2/m$ or $R\bar{3}m$) & Li-containing rock-salt $[Li_xTM_{1-x}]O$ ($Fm\bar{3}m$) nano-domains in our electrodes, which agrees quite well with the analysis of activation process (layered-to-rock-salt transition) of L1.20 electrode during the first 100 cycles in **Figure S7 & S8**. The disordered rock-salt-type phase can block the lithium diffusion pathway to some extent and thus results in a worsening electrochemical performance.⁴ (page 14-15)

Taken together, thermodynamically driven formation of the Li-poor spinel phase (AB_2O_4 , $Fd\bar{3}m$) is finally observed after ultra-long cycling time, which arises from irreversible release of lithium and oxygen from layered crystal lattice and is supposed to be located at the surface (see **Figure 4**). Unfortunately, the accumulation of layered & rock-salt-type & spinel domains with the large lattice mismatch induces a much higher diffusion barrier for Li ions, compared to the stage II, and thereby a considerably capacity loss at stage III in **Figure 2(c)**. In addition, the capacity fading is probably also a consequence of the slow dissolution of manganese (III) ions in the spinel phase ($2Mn^{3+} \rightarrow Mn^{4+} + Mn^{2+}$), the lithium dendrite and SEI growth in the coin cell, and electrolyte evaporation.⁵⁻⁸ Most importantly, such long-term cycling battery test could give valuable information about the kinetic behavior of the phase transition from a Li-rich layered $Li[Li_xTM_{1-x}]O_2$ ($C2/m$ and/or $R\bar{3}m$) to a partially disordered Li-containing rock-salt $[Li_xTM_{1-x}]O$ ($Fm\bar{3}m$) and then to a Li-poor spinel $(Li_xTM_{1-x})[TM_2]O_4$ ($Fd\bar{3}m$) structure. (page 17)

Comment #2: Furthermore, although it is interesting to make the comparison between phase evolution at high temperature synthesis conditions and that at room temperature

electrochemical cycling conditions, I am not sure the authors made it clear the fundamental connection in the very different temperature scales. At high temperature, many kinetic barriers can be easily overcome, such as the oxygen vacancy diffusion barrier or transition metal migration barrier, while at room temperature, these barriers could largely limit or even forbid a thermodynamic phase transformation. Thus, it is not clear why the high temperature phase evolution can give insights to the room temperature one here. In my opinion, it will be more valuable to focus on the differences between different temperature scales, and discuss more about the unique electrochemical structural evolution at room temperature.

Answer #2: We thank the reviewer for the insightful comments, we agree with these comments. Our experimental results demonstrate that the high-temperature reaction of spinel oxide with Li/O species is thermodynamically favored in the ambient atmosphere, and the formation of Li-containing rock-salt phase is a transition state during high-temperature lithiation reaction. The activation energy barrier from Li-containing rock-salt intermediate to Li-rich layered oxide can easily be transcended via a fast-heating rate, but such activation barrier could impede the thermodynamic phase transformation at room temperature. This could help to explain why the TM cation migration into the octahedral positions of Li-containing rock-salt-type/layered phase in the near-surface region, rather than the tetrahedral positions of cubic spinel phase ($Fd\bar{3}m$), is so often observed in Li-rich layered cathode materials during a short-term to midterm (e.g., 100 cycles) cycling (kinetic control) and why Li-rich layered oxides have the tendency to form Li-containing rock-salt-type & Li-poor spinel coherent phases after a certain amount of Li/O loss during prolonged cycling (> 500 cycles) at room temperature (i.e. thermodynamically driven).^{1,9-11} Another significant intrinsic connection between formation and degradation of Li-rich layered oxides is how and where Li/O species are incorporated/released into/from the crystal structure with ccp oxygen sublattice during synthesis/cycling, see the discussion in our updated manuscript.

A comprehensive and nuanced analysis of the subject, i.e. the fundamental connection

between electrochemical property and phase evolution at different temperature scales, was done in our previous work¹². We demonstrate that, by increasing the calcination temperature, lithium and oxygen gradually enter into the intermediate cubic spinel/rock-salt-type host framework of the Li-free precursor and continuously cause the formation of a layered phase. Compared to the temperature-dependent experiments, the advantage of Li-concentration-dependent experiments (**Figure 1** in the revised manuscript) is that most of the Li from the Li source is undoubtedly inserted into the host architecture during high-temperature lithiation reaction (e.g., 850 °C), rather than attached on the surface of crystallites at a lower heating temperature (e.g., < 800 °C). The structural evolution at different temperature scales for a relatively long heating time¹² matches precisely with the structural and electronic analysis of thermally stable $\text{Li}_x\text{Ni}_{0.2}\text{Mn}_{0.6}\text{O}_y$ oxides (**Figure 1**) and the structural evolution of Li-free spinel with incorporation of Li/O during slow heating (**Figure 5a**).

Thermodynamically, different temperature scales are strongly correlated to various time scales. As temperature decreases, the time needed to reach equilibrium increases exponentially, this is why the thermodynamically driven formation of spinel (AB_2O_4 , $Fd\bar{3}m$) phase during degradation of Li-rich oxides is so difficult to observe in the cycled electrode within a limited cycling time at room temperature. The low temperature or fast heating process will take the system far away from equilibrium. During synthesis of layered $\text{Li}[\text{Li}_{1.2}\text{Ni}_{0.2}\text{Mn}_{0.6}]\text{O}_2$ oxide under rapid-heating conditions, the cubic spinel $\text{Mn}[\text{Ni}_{0.75}\text{Mn}_{1.25}]\text{O}_4$ phase (L0.00) experiences a transformation from a Li-free spinel ($Fd\bar{3}m$) to a fully disordered Li-containing rock-salt ($Fm\bar{3}m$) and finally to a Li-rich layered structure ($C2/m$) with the incorporation of lithium and oxygen. As a gradual Li/O loss during long-term cycling occurs, the Li-rich layered $\text{Li}[\text{Li}_{1.2}\text{Ni}_{0.2}\text{Mn}_{0.6}]\text{O}_2$ cathode tends to transform back to the ‘Li-poor’ spinel phase ($Fd\bar{3}m$), but this transition is limited by the generation of the Li-containing rock-salt phase ($Fm\bar{3}m$). Such an inversed relationship between formation pathway and degradation process of LMLOs is not a coincidence but intrinsic due to

the fact that both occur under non-equilibrium conditions driven by Li/O concentration gradient.

Therefore, all these consistent results provide objective evidence of correlation between the electrochemical performances and the thermal stability of the Li-rich oxides, which could inspire others to develop a convenient and productive methodology for mitigating the capacity loss and voltage decay without performing time-consuming cycling.

Reference:

1. Zheng, J. *et al.* Structural and chemical evolution of Li- and Mn-rich layered cathode material. *Chem. Mater.* **27**, 1381–1390 (2015).
2. Tackett, R. *et al.* Magnetodielectric coupling in Mn_3O_4 . *Phys. Rev. B* **76**, 024409 (2007).
3. Kim, M. *et al.* Mapping the Magnetostructural Quantum Phases of Mn_3O_4 . *Phys. Rev. Lett.* **104**, 136402 (2010).
4. Xu, G. L. *et al.* Building ultraconformal protective layers on both secondary and primary particles of layered lithium transition metal oxide cathodes. *Nat. Energy* (2019). doi:10.1038/s41560-019-0387-1
5. Li, W. *et al.* The synergetic effect of lithium polysulfide and lithium nitrate to prevent lithium dendrite growth. *Nat. Commun.* **6**, (2015).
6. Pieczonka, N. P. W. *et al.* Understanding transition-metal dissolution behavior in $\text{LiNi}_{0.5}\text{Mn}_{1.5}\text{O}_4$ high-voltage spinel for lithium ion batteries. *J. Phys. Chem. C* (2013). doi:10.1021/jp405158m
7. McArthur, M. A., Trussler, S. & Dahn, J. R. In Situ Investigations of SEI Layer Growth on Electrode Materials for Lithium-Ion Batteries Using Spectroscopic Ellipsometry. *J. Electrochem. Soc.* (2011). doi:10.1149/2.004203jes
8. Ramadass, P., Haran, B., White, R. & Popov, B. N. Capacity fade of Sony 18650 cells cycled at elevated temperatures: Part II. Capacity fade analysis. *J. Power Sources* (2002). doi:10.1016/S0378-7753(02)00473-1
9. Yan, P. *et al.* Injection of oxygen vacancies in the bulk lattice of layered cathodes. *Nat. Nanotechnol.* (2019). doi:10.1038/s41565-019-0428-8
10. Xu, B., Fell, C. R., Chi, M. & Meng, Y. S. Identifying surface structural changes in layered Li-excess nickel manganese oxides in high voltage lithium ion batteries: A joint experimental and theoretical study. *Energy Environ. Sci.* (2011). doi:10.1039/c1ee01131f
11. Zheng, J. *et al.* Mitigating voltage fade in cathode materials by improving the atomic level uniformity of elemental distribution. *Nano Lett.* **14**, 2628–2635 (2014).
12. Hua, W. *et al.* Lithium/Oxygen Incorporation and Microstructural Evolution during Synthesis of Li-Rich Layered $\text{Li}[\text{Li}_{0.2}\text{Ni}_{0.2}\text{Mn}_{0.6}]\text{O}_2$ Oxides. *Adv. Energy Mater.* **8**, 1803094 (2019).

REVIEWERS' COMMENTS:

Reviewer #1 (Remarks to the Author):

The authors made substantial improvements to their results and discussions based on reviewers' comments. Therefore, this reviewer recommends the publication of this revised manuscript in Nature Communications.

Reviewer #2 (Remarks to the Author):

The authors have fully addressed the questions/comments that I had in last review. In addition, they also added the Bragg Coherent Diffraction Imaging data, which is a good proof that the degradation of the Li-rich cathode is surface-bulk rate limited.

The authors also explained better in the revised version why a deep analysis on the heating/synthesis process is valuable to understand the degradation of Li-rich cathode. The synthesis and electrochemical degradation processes take place at different temperature scales. Yet the electrochemical degradation also takes place in a much longer time scale than synthesis and the electrochemical potential is also a strong driving force (or source of fluctuation) that can quite effectively initiate the migration of ions at low temperature, similar to the heat-induced ionic migration at higher temperatures. So it is indeed necessary and intriguing to understand the degradation from a new angle, the synthesis process.

Such new approach is novel and rather rarely seen in previous literature. I keep my recommendation in last review and believe this manuscript should be accepted for publication in Nature communications.

Reviewer #3 (Remarks to the Author):

The revised version is much improved, and the discussion about the relationship between the high temperature phase evolution and the room temperature electrochemical phase evolution is satisfactory in terms of thermodynamics, kinetics and structural analysis. The importance of these results finally emerges with the discussion. I thus recommend the manuscript to be considered for publication on Nature Communications.

Point-by-point response to the reviewers' comments (in blue)

Reviewer #1:

The authors made substantial improvements to their results and discussions based on reviewers' comments. Therefore, this reviewer recommends the publication of this revised manuscript in Nature Communications.

Response: We highly appreciate the reviewer's positive comments.

Reviewer #2:

The authors have fully addressed the questions/comments that I had in last review. In addition, they also added the Bragg Coherent Diffraction Imaging data, which is a good proof that the degradation of the Li-rich cathode is surface-bulk rate limited.

The authors also explained better in the revised version why a deep analysis on the heating/synthesis process is valuable to understand the degradation of Li-rich cathode. The synthesis and electrochemical degradation processes take place at different temperature scales. Yet the electrochemical degradation also takes place in a much longer time scale than synthesis and the electrochemical potential is also a strong driving force (or source of fluctuation) that can quite effectively initiate the migration of ions at low temperature, similar to the heat-induced ionic migration at higher temperatures. So it is indeed necessary and intriguing to understand the degradation from a new angle, the synthesis process.

Response: We highly appreciate the reviewer's positive and insightful comments.

Reviewer #3:

The revised version is much improved, and the discussion about the relationship between the high temperature phase evolution and the room temperature electrochemical phase evolution is satisfactory in terms of thermodynamics, kinetics and structural analysis. The importance of these results finally emerges with the discussion. I thus recommend the manuscript to be considered for publication on Nature Communications.

Response: We highly appreciate the reviewer's positive comments.